# INVERTIBLE MANIFOLD LEARNING FOR DIMENSION REDUCTION

## ABSTRACT

It is widely believed that a dimension reduction (DR) process drops information inevitably in most practical scenarios. Thus, most methods try to preserve some essential information of data after DR, as well as manifold based DR methods. However, they usually fail to yield satisfying results, especially in high-dimensional cases. In the context of manifold learning, we think that a good low-dimensional representation should preserve the topological and geometric properties of data manifolds, which involve exactly the entire information of the data manifolds. In this paper, we define the problem of information-lossless NLDR with the manifold assumption and propose a novel two-stage NLDR method, called invertible manifold learning (*inv-ML*), to tackle this problem. A *local isometry* constraint of preserving local geometry is applied under this assumption in *inv-ML*. Firstly, a homeomorphic *sparse coordinate transformation* is learned to find the low-dimensional representation without losing topological information. Secondly, a *linear compression* is performed on the learned sparse coding, with the trade-off between the target dimension and the incurred information loss. Experiments are conducted on seven datasets with a neural network implementation of *inv-ML*, called *i-ML-Enc*, which demonstrate that the proposed *inv-ML* not only achieves invertible NLDR in comparison with typical existing methods but also reveals the characteristics of the learned manifolds through linear interpolation in latent space. Moreover, we find that the reliability of tangent space approximated by the local neighborhood on real-world datasets is key to the success of manifold based DR algorithms. The code will be made available soon.

## 1 INTRODUCTION

In real-world scenarios, it is widely believed that the loss of data information is inevitable after dimension reduction (DR), though the goal of DR is to preserve as much information as possible in the low-dimensional space. In the case of linear DR, compressed sensing (Donoho, 2006) breaks this common sense with practical sparse conditions of the given data. In the case of nonlinear dimension reduction (NLDR), however, it has not been clearly discussed, e.g. what is the structure within data and how to maintain these structures after NLDR? From the perspective of manifold learning, the *manifold assumption* is widely adopted, but classical manifold based DR methods usually fail to yield good results in the many practical case. Therefore, what is the gap between theoretical and real-world applications of manifold based DR? Here, we give the first detailed discussion of these two problems in the context of manifold learning. We think that a good low-dimensional representation should preserve the topology and geometry of input data, which require the NLDR transformation to be homeomorphic. Thus, we propose an invertible NLDR process, called *inv-ML*, combining *sparse coordinate transformation* and *local isometry* constraint which preserve the property of topology and geometry, to explain the information-lossless NLDR in manifold learning theoretically. We instantiate *inv-ML* as a neural network called *i-ML-Enc* via a cascade of equidimensional layers and a linear transform layer. Sufficient experiments are conduct to validate invertible NLDR abilities of *i-ML-Enc* and analyze learned representations to reveal inherent difficulties of classical manifold learning.

**Topology preserving dimension reduction.** To start, we first make out the theoretical definition of information-lossless DR on a manifold. The topological property is what is invariant under a homeomorphism, and thus what we want to achieve is to construct a homeomorphism for dimension

reduction, removing the redundant dimensions while preserving invariant topology. To be more specific, $f : \mathcal{M}_0^d \to \mathbb{R}^m$ is a smooth mapping of a differential manifold into another, and if $f$ is a homeomorphism of $\mathcal{M}_0^d$ into $\mathcal{M}_1^d = f(\mathcal{M}_0^d) \subset \mathbb{R}^m$, we call $f$ is an embedding of $\mathcal{M}_0^d$ into $\mathbb{R}^m$. Assume that the data set $\mathcal{X} = \{\boldsymbol{x}_j | 1 \leq j \leq n\}$ sampled from the compact manifold $\mathcal{M}_1^d \subset \mathbb{R}^m$ which we call the data manifold and is homeomorphic to $\mathcal{M}_0^d$. For the sample points we get are represented in the coordinate after inclusion mapping $i_1$, we can only regard them as points from Euclidean space $\mathbb{R}^m$ without any prior knowledge, and learn to approximate the data manifold in the latent space $Z$. According to the Whitney Embedding Theorem (Seshadri & Verma, 2016), $\mathcal{M}_0^d$ is can be embedded smoothly into $\mathbb{R}^{2d}$ by a homeomorphism $g$. Rather than to find the $f^{-1} : \mathcal{M}_1^d \to \mathcal{M}_0^d$, our goal is to seek a smooth map $h : \mathcal{M}_1^d \to \mathbb{R}^s \subset \mathbb{R}^{2d}$, where $h = g \circ f^{-1}$ is a homeomorphism of $\mathcal{M}_1^d$ into $\mathcal{M}_2^d = h(\mathcal{M}_1^d)$ and $d \leq s \leq 2d \ll m$, and thus the $dim(h(\mathcal{X})) = s$, which achieves the DR while preserving the topology. Owing to the homeomorphism $h$ we seek as a DR mapping, the data manifold $\mathcal{M}_1^d$ is reconstructible via $\mathcal{M}_1^d = h^{-1} \circ h(\mathcal{M}_1^d)$, by which we mean $h$ a topology preserving DR as well as information-lossless DR.

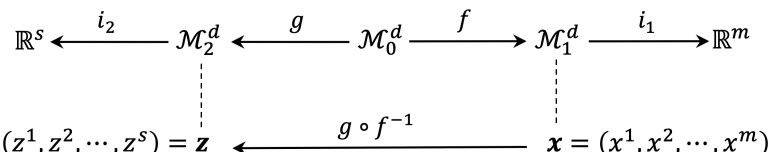

Figure 1: Illustration of the process of NLDR. The dash line links $\mathcal{M}_1^d$ and $\boldsymbol{x}$ means $\boldsymbol{x}$ is sampled from $\mathcal{M}_1^d$, and it is represented in the Euclidean space $\mathbb{R}^m$ after an inclusion mapping $i_i$. We aim to approximate $\mathcal{M}_1^d$ from the observed sample $\boldsymbol{x}$. For the topology preserving dimension reduction methods, it aims to find a homeomorphism $g \circ f^{-1}$ to map $\boldsymbol{x}$ into $\boldsymbol{z}$ which is embedded in $\mathbb{R}^s$.

**Geometry preserving dimension reduction.** While the topology of the data manifold $\mathcal{M}_1^d$ can be preserved by the homeomorphism $h$ discussed above, it may distort the geometry. To preserve the local geometry of the data manifold, the map should be isometric on the tangent space $\mathcal{T}_p \mathcal{M}_1^d$ for every $p \in \mathcal{M}_1^d$, indicating that $d_{\mathcal{M}_1^d}(u, v) = d_{\mathcal{M}_2^d}(h(u), h(v)), \forall u, v \in \mathcal{T}_p \mathcal{M}_1^d$. By Nash's Embedding Theorem (Nash, 1956), any smooth manifold of class $C^k$ with $k \geq 3$ and dimension $d$ can be embedded isometrically in the Euclidean space $\mathbb{R}^s$ with $s$ polynomial in $d$.

**Noise perturbation.** In the real-world scenarios, sample points are not lied on the ideal manifold strictly due to the limitation of sampling, e.g. non-uniform sampling noises. When the DR method is very robust to the noise, it is reasonable to ignore the effects of the noise and learn the representation $Z$ from the given data. Therefore, the intrinsic dimension of $\mathcal{X}$ is approximate to $d$, resulting in the lowest isometric embedding dimension is larger than $s$.

## 2 RELATED WORK

**Manifold learning.** Most classical linear or nonlinear DR methods aim to preserve the geometric properties of manifolds. The Isomap (Tenenbaum et al., 2000) based methods aim to preserve the global metric between every pair of sample points. For example, McQueen et al. (2016) can be regarded as such methods based on the push-forward Riemannian metric. For the other aspect, LLE (Roweis & Saul, 2000) based methods try to preserve local geometry after DR, whose derivatives like LTSA (Zhang & Zha, 2004), MLLE (Zhang & Wang, 2007), etc. have been widely used but usually fail in the high-dimensional case. Recently, based on local properties of manifolds, MLDL (Li et al., 2020) was proposed as a robust NLDR method implemented by a neural network, preserving the local geometry but abandoning the retention of topology. In contrast, our method takes the preservation of both geometry and topology into consideration, trying to maintain these properties of manifolds even in cases of excessive dimension reduction when the target dimension $s'$ is smaller than $s$.

**Invertible model.** From AutoEncoder (AE) (Hinton & Salakhutdinov, 2006), the fundamental neural network based model, having achieved DR and cut information loss by minimizing the reconstruction loss, some AE based generative models like VAE (Kingma & Welling, 2014) and manifold-based NLDR models like TopoAE (Moor et al., 2020) has emerged. These methods cannot avoid information loss after NLDR, and thus, some invertible models consist of a series of

equidimensional layers have been proposed, some of which aim to generate samples by density estimation through layers (Dinh et al., 2015) (Dinh et al., 2017) (Behrmann et al., 2019), and the other of which are established for other targets, e.g. validating the mutual information bottleneck (Jacobsen et al., 2018). Different from methods mentioned above, our proposed *i-ML-Enc* is a neural network based encoder, with NLDR as well as maintaining structures of raw data points based on manifold assumption via a series of equidimensional layers.

**Compressed sensing.** The JohnsonLindenstrauss Theorem (Johnson & Lindenstrauss, 1984) provides the lower bound of target dimension for linear DR with the pairwise distance loss. Given a small constant $\epsilon \in (0, 1)$ and $n$ samples $\{\boldsymbol{x}_i\}_{i=1}^n$ in $\mathbb{R}^m$, a linear projection $W : \mathbb{R}^m \to \mathbb{R}^s, s > O(\frac{logm}{\epsilon^2})$ can be found, which embeds samples into a $s$-dimensional space with $(1 + \epsilon)$ distortion of any sample pairs $(\boldsymbol{x}_i, \boldsymbol{x}_j)$. It adopts a prior assumption that the given samples in high-dimensional space have a relevant low-dimensional structure constraint which can be maintained by keeping the pairwise distance. Further, compressed sensing (CS) provides strict sparse conditions of linear DR with great probability to recover the compressed signal, which usually cooperates with sparse dictionary learning (Hawe et al., 2013). The core of CS is Restricted Isometry Property (RIP) condition, which reads

$$(1 - \epsilon)\|\boldsymbol{x}_1 - \boldsymbol{x}_2\|^2 \leq \|W(\boldsymbol{x}_1 - \boldsymbol{x}_2)\|^2 \leq (1 + \epsilon)\|\boldsymbol{x}_1 - \boldsymbol{x}_2\|^2, \tag{1}$$

where $\epsilon \in (0, 1)$ is a rather small constant and $W$ is a linear measurement of signal $\boldsymbol{x}_1$ and $\boldsymbol{x}_2$. Given a signal $\boldsymbol{x} \in \mathbb{R}^m$ with $s$-sparse representation $\alpha = \Phi x$ on an $m$-dimensional orthogonal basis $\Phi$, $\alpha$ can be recovered from the linear measurement $\boldsymbol{y} = W\alpha$ with great probability by the sparse optimization if $W_{m \times s}$ satisfies the RIP condition: $\arg \min_{\tilde{\alpha}} \|\tilde{\alpha}\|_0, \ s.t. \ y = W\tilde{\alpha}$. The linear measurement is rewritten as $\boldsymbol{y} = \Psi\Phi\alpha = \Psi\boldsymbol{x}$ where $\Psi$ is a low-dimensional orthogonal basis and $\Phi$ can be found by the nonlinear dictionary learning. Some reconstructible CS-based NLDR methods (Wei et al., 2015) (Wei et al., 2019) are proposed, which are achieved by preserving local geometry on AE-based networks, but usually with unsatisfying embedding qualities.

## 3 PROPOSED METHOD

We will specifically discuss the proposed two-stage invertible NLDR process *inv-ML* as the first stage in Sec 3.1, in which a $s$-dimensional representation is learned by a homeomorphism transformation while keeping all topological and geometric structure of the data manifold; then give applicable conditions in real-world scenarios as the second stage in Sec 3.2, in which the dimension is further compressed to $s'$. We instantiate the proposed *inv-ML* as a neural network *i-ML-Enc* in Sec 3.3.

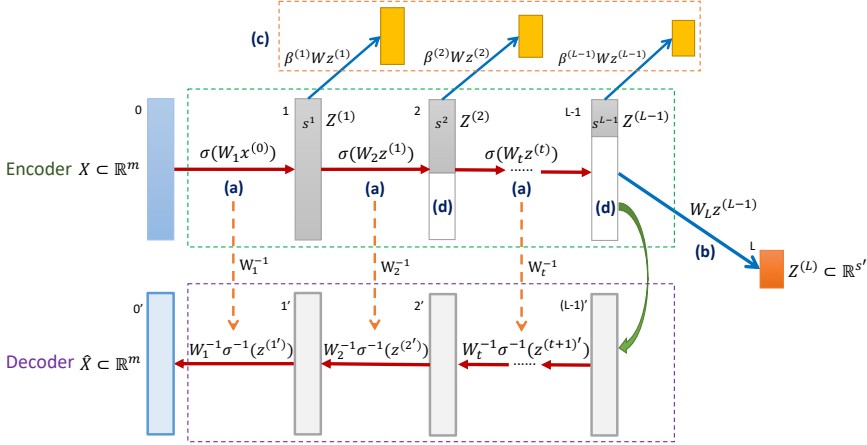

Figure 2: The network structure for the proposed implementation *i-ML-Enc*. The first $L - 1$ layers equidimensional mapping in the green dash box are the first stage which achieves $s$-sparse, and they have an inverse process in the purple dash box. (a) is a layer of nonlinear homeomorphism transformation (red arrow). (b) linearly transforms (blue arrow) $s$-sparse representation in $\mathbb{R}^m$ into $\mathbb{R}^{s'}$ as the second stage. (c) are the *extra heads* by linear transformations. (d) indicates the padding zeros of the $l$-th layer to force $d^{(l)}$-sparse.

### 3.1 TOPOLOGY AND GEOMETRY PRESERVATION

**Canonical embedding for homeomorphism.** To seek the smooth homeomorphism $h$, we turn to the theorem of local canonical form of immersion (Mei, 2013). Let $f : \mathcal{M} \to \mathcal{N}$ an immersion, and for any $p \in \mathcal{M}$, there exist local coordinate systems $(U, \phi)$ around $p$ and $(V, \psi)$ around $f(p)$ such that $\psi \circ f \circ \phi^{-1} : \phi(U) \to \psi(V)$ is a canonical embedding, which reads

$$\psi \circ f \circ \phi^{-1}(x^1, x^2, \cdots, x^d) = (x^1, x^2, \cdots, x^d, 0, 0, \cdots, 0). \tag{2}$$

In our case, let $\mathcal{M} = \mathcal{M}_2^d$, and $\mathcal{N} = \mathcal{M}_1^d$, any point $\boldsymbol{z} = (z^1, z^2, \cdots, z^s) \in \mathcal{M}_1^d \subset \mathbb{R}^s$ can be mapped to a point in $\mathbb{R}^m$ by the canonical embedding

$$\psi \circ h^{-1}(z^1, z^2, \cdots, z^s) = (z^1, z^2, \cdots, z^s, 0, 0, \cdots, 0). \tag{3}$$

For the point $\boldsymbol{z}$ is regarded as a point in $\mathbb{R}^s$, $\phi = \mathbb{I}$ is an identity mapping, and for $h = g \circ f^{-1}$ is a homeomorphism, $h^{-1}$ is continuous. The Eq. (3) can be written as

$$\begin{aligned}(z^1, z^2, \cdots, z^s) &= h \circ \psi^{-1}(z^1, z^2, \cdots, z^s, 0, 0, \cdots, 0) \\ &= h(x^1, x^2, \cdots, x^m). \end{aligned} \tag{4}$$

Therefore, to reduce $dim(\mathcal{X}) = m$ to $s$, we can decompose $h$ into $\psi$ and $h \circ \psi^{-1}$, by firstly finding a homeomorphic coordinate transformation $\psi$ to map $\boldsymbol{x} = (x^1, x^2, \cdots, x^m)$ into $\psi(\boldsymbol{x}) = (z^1, z^2, \cdots, z^s, 0, 0, \cdots, 0)$, which is called a *sparse coordinate transformation*, and $h \circ \psi^{-1}$ can be easily obtained by Eq. (3). We denote $h \circ \psi^{-1}$ by $h_0$ and call it a *sparse compression*. The theorem holds for any manifold, while in our case, we aims to find the mapping of $\mathcal{X} \subset \mathbb{R}^m$ into $\mathbb{R}^s$, so the local coordinate systems can be extended to the whole space of $\mathbb{R}^m$.

**Local isometry constraint.** The prior local isometry constraint is applied under the manifold assumption, which aims to preserve distances (or some other metrics) locally so that $d_{\mathcal{M}_1^d}(u, v) = d_{\mathcal{M}_2^d}(h(u), h(v)), \forall u, v \in \mathcal{T}_p \mathcal{M}_1^d$.

### 3.2 LINEAR COMPRESSION

With the former discussed method, manifold-based NLDR can be achieved with topology and geometry preserved, i.e. $s$-sparse representation in $\mathbb{R}^m$. However, the target dimension $s'$ may be even less than $s$, further compression can be performed through the *linear compression* $h_0' : \mathbb{R}^m \to \mathbb{R}^{s'}$ instead of *sparse compression*, where $h_0'(\boldsymbol{z}) = W_{m \times s'} \boldsymbol{z}$, with minor information loss. In general, the *sparse compression* is a particular case of *linear compression* with $h_0(\boldsymbol{z}) = h_0'(\boldsymbol{z}) = \Lambda \boldsymbol{z}$, where $\Lambda = (\delta_{i,j})_{m \times s}$ and $\delta_{i,j}$ is the Kronecker delta. We discusses the information loss caused by a linear compression under different target dimensions $s'$ as following cases.

**Ideal case.** In the case of $d \leq s \leq s'$, based on compressed sensing, we can reconstruct the raw input data after NLDR process without loss of any information by solving the sparse optimization problem mentioned in Sec. 2 when the transformation matrix $W_{m \times s'}$ has full rank of the column. In the case of $d \leq s' < s$, it is inevitable to drop the topological properties because the two spaces before and after NLDR are not homeomorphic, and it is reduced to local geometry preservation by LIS constraint. However, in the case of $s' \leq d < s$, both topological and geometric information is lost to varying degrees. Therefore, we can only try to retain as much geometric structure as possible.

**Practical case.** In real-world scenarios, the target dimension $s'$ is usually lower than $s$, even lower than $d$. Meanwhile, the data sampling rate is quite low, and the clustering effect is extremely significant, indicating that it is possible to approximate $\mathcal{M}_1$ by low-dimensional hyperplane in the Euclidean space. In the case of $s' < s$, we can retain as the prior Euclidean topological structure as additional topological information of raw data points. It is reduced to replace the global topology with some relative structures between each cluster.

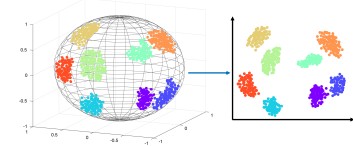

Figure 3: Sparsity and clustering effect.

### 3.3 NETWORK FOR IMPLEMENTATION

Based on Sec 3.1 and Sec 3.2, we propose a neural network *i-ML-Enc* which achieves two-stage NLDR preserving both topology and geometry, as shown in Fig. 2. In this section, we will introduce the function of network structures and loss functions respectively, including the orthogonal loss, padding loss and *extra heads* for the first stage, and the LIS loss, push-away loss for the second stage.

**Cascade of homeomorphisms.** Since the *sparse coordinate transformation* $\psi$ (and its inverse) can be highly nonlinear and complex, we decompose it into a cascade of $L-1$ isometric homeomorphisms $\psi = \psi^{(L-1)} \circ \cdots \circ \psi^{(2)} \circ \psi^{(1)}$, which can be achieved by $L-1$ equidimensional network layers. For each $\psi^{(l)}$, it is a *sparse coordinate transformation*, where $\psi^l(z^{1,(l)}, z^{2,(l)}, \cdots, z^{s_l,(l)}, 0, \cdots, 0) = (z^{1,(l+1)}, z^{2,(l+1)}, \cdots, z^{s_{l+1},(l+1)}, 0, \cdots, 0)$ with $s_{l+1} < s_l$ and $s_{L-1} = s$. The layer-wise transformation $Z^{(l+1)} = \psi^{(l)}(Z^{(l)})$ and its inverse can be written as

$$Z^{(l+1)} = \sigma(W_l X^{(l)}), \ Z^{(l)'} = W_l^{-1}(\sigma^{-1}(Z^{(l+1)'})), \tag{5}$$

in which $W_l$ is the $l$-th weight matrix of the neural network to be learned, and $\sigma(.)$ is a nonlinear activation. The bias term is removed here to facilitate its simple inverse structure.

**Orthogonal loss.** Each layer-wise transformation is thought to be a homeomorphism between $Z^{(l)}$ and $Z^{(l+1)}$, and we want it to be a nearly isometric. We force each $W_l$ to be an orthogonal matrix, which allows simple calculation of the inverse of $W_l$. Based on RIP condition, the orthogonal constraint of the weight matrix in the first $L-1$ layers can be obtained as

$$L_{orth} = \sum_{l=1}^{L-1} \alpha^{(l)} \rho(W_l^T W_l - I), \tag{6}$$

where $\{\alpha^{(l)}\}$ are the loss weights. Notice that $\rho(W) = \sup_{z \in \mathbb{R}^m, z \neq 0} \frac{|Wz|}{|z|}$ is the spectral norm of $W$, and the loss term can be written as $\rho(W_l^T W_l - I) = \sup_{z \in \mathbb{R}^m, z \neq 0} |\frac{|Wz|}{|z|}|$ which is equivalent to RIP condition in Eq. (1).

**Padding loss.** To force sparsity from the second to $(L-1)$-th layers, we add a zero padding loss to each of these layers. For the $l$-th layer whose target dimension is $s_l$, pad the last $m - s_l$ elements of $z^{(l+1)}$ with zeros and panish these elements with $L_1$ norm loss:

$$L_{pad} = \sum_{l=2}^{L-1} \beta^{(l)} \sum_{i=s^{(l)}}^{m} |z_i^{(l+1)}|, \tag{7}$$

where $\{\beta^{(l)}\}$ are loss weights. The target dimension $s_l$ can be set heuristically.

**Linear transformation head.** We use the linear transformation head to achieve the linear compression step in our NLDR process, which is a transformation between the orthogonal basis of high dimension and lower dimension. Thus, we apply the row orthogonal constraint to $W_L$.

**LIS loss.** Since the linear DR is applied at the end of the NLDR process, we apply *locally isometric smoothness* (LIS) constraint (Li et al., 2020) to preserve the local geometric properties. Take the LIS loss in the $l$-th layer as an example:

$$L_{LIS} = \sum_{i=1}^{n} \sum_{j \in \mathcal{N}_i^k} \left\| d_X(x_i, x_j) - d_Z(z_i^{(l)}, z_j^{(l)}) \right\|, \tag{8}$$

where $\mathcal{N}_i^k$ is a set of $x_i$'s $k$-nearest neighborhood in the input space, and $d_X$ and $d_Z$ are the distance of the input and the latent space, which can be approximated by Euclidean distance in local open sets.

**Push-away loss.** In the real case discussed in Sec 3.2, the latent space of the $(L-1)$-th layer can approximately to be a hyperplane in Euclidean space, so that we introduce push-away loss to repel the non-adjacent sample points of each $x_i$ in its $B$-radius neighbourhood in the latent space. It deflates the manifold locally when acting together with $L_{LIS}$ in the linear DR. Similarly, $L_{push}$ is applied after the linear transformation in the $l$-th layer:

$$L_{push} = -\sum_{i=1}^{n} \sum_{j \in \mathcal{N}_i^k} \mathbf{1}_{d_Z(z_i^{(l)}, z_j^{(l)}) < B} \log \left( 1 + d_Z(z_i^{(l)}, z_j^{(l)}) \right), \tag{9}$$

where $\mathbf{1}(.) \in \{0, 1\}$ is the indicator function for the bound of $B$.

**Extra heads.** In order to force the first $L - 1$ layers of the network to achieve NLDR gradually, we introduce auxiliary DR branchs, called *extra head*, at layers from the second to the $(L - 1)$-th. The structure of each *extra head* is same as the linear transformation head and will be discarded after training. $L_{extra}$ is written as

$$L_{extra} = \sum_{l=1}^{L-1} \gamma^{(l)}(L_{LIS} + \mu^{(l)} L_{push}), \tag{10}$$

where $\{\gamma^{(l)}\}$ and $\{\mu^{(l)}\}$ are loss weights which can be set based on $\{s_l\}$.

**Inverse process.** The inverse process is the decoder directly obtained by the first $L - 1$ layers of the encoder given by Eq. (5), which does not involved in the training process. When the target dimension $s'$ is equal to $s$, the inverse of the layer-$L$ can be solved by some existing methods such as compressed sensing or eigenvalue decomposition.

## 4 EXPERIMENT

In this section, we first evaluate the proposed invertible NLDR achieved by *i-ML-Enc* in Sec 4.1, then investigate the property of data manifolds with *i-ML-Enc* in Sec 4.2. The properties of *i-ML-Enc* are further studied in Sec 4.3. We carry out experiments on **seven datasets**: (i) Swiss roll (Pedregosa et al., 2011), (ii) Spheres (Moor et al., 2020) and Half Spheres, (iii) USPS (Hull, 1994), (iv) MNIST (LeCun et al., 1998), (v) KMNIST (Clanuwat et al., 2018), (vi) FMNIST (Xiao et al., 2017), (vii) COIL-20 (Nene et al., 1996b). The implementation is based on the PyTorch 1.3.0 library running on NVIDIA v100 GPU. The following settings of *i-ML-Enc* are used for all datasets: LeakyReLU with $\alpha = 0.1$; Adam optimizer (Kingma & Ba, 2015) with learning rate $lr = 0.001$ for 8000 epochs; the local neighborhood is determined by kNN with $k = 15$; $L$ layers neural network as shown in Fig. 2.

### 4.1 METHODS COMPARISON

To verify the invertible NLDR ability of *i-ML-Enc* and analyze different cases of NLDR, we compare it with several typical methods in NLDR and inverse scenarios on both synthetic (Swiss roll, Spheres and Half Spheres) and real-world datasets (USPS, MNIST, FMNIST and COIL-20). **Six methods for manifold learning**: MLLE (Zhang & Wang, 2007), t-SNE (Maaten & Hinton, 2008) and ML-Enc (Li et al., 2020) are compared for NLDR; three AE-based methods VAE (Kingma & Welling, 2014), TopoAE (Moor et al., 2020) and ML-AE (Li et al., 2020) are compared for reconstructible manifold learning. **Three methods for inverse models**: INN (Nguyen et al., 2019), i-RevNet (Jacobsen et al., 2018), and i-ResNet (Behrmann et al., 2019) are compared for bijective property. Among them, i-RevNet and i-ResNet are supervised algorithms while the rest are unsupervised. For a fair comparison in this experiment, we adopt 8 layers neural network for all the network-based methods except i-RevNet and i-ResNet. **Hyperparameter** values of *i-ML-Enc* and configurations of these datasets such as the input and target dimension are provided in **Appendix A.2**.

Table 1: Comparison in representation and invertible quality on MNIST datasets

| Dataset | Algorithm | RMSE | MNE | Trust | Cont | $K$min | $K$max | $l$-MSE | Acc |
|---------|-----------|------|-----|-------|------|--------|--------|---------|-----|
| MNIST | MLLE | - | - | 0.6709 | 0.6573 | 1.873 | 6.7e+9 | 36.80 | 0.8341 |
| | t-SNE | - | - | 0.9896 | 0.9886 | 5.156 | 324.9 | 48.07 | 0.9246 |
| | ML-Enc | - | - | 0.9862 | **0.9927** | 1.761 | 58.91 | 18.98 | 0.9326 |
| | VAE | 0.5263 | 33.17 | 0.9712 | 0.9703 | 5.837 | 130.5 | 22.79 | 0.8652 |
| | TopoAE | 0.5178 | 31.45 | **0.9915** | 0.9878 | 4.943 | 265.3 | 24.98 | 0.8993 |
| | ML-AE | 0.4012 | 16.84 | 0.9893 | 0.9926 | **1.704** | **57.48** | 19.05 | **0.9340** |
| | i-ML-Enc (L) | **0.0457** | **0.5085** | 0.9906 | 0.9912 | 2.033 | 60.14 | **18.16** | 0.9316 |
| | INN | 0.0615 | 0.5384 | 0.9851 | 0.9823 | 1.875 | 22.38 | 7.494 | 0.9176 |
| | i-RevNet | 0.0443 | **0.4679** | 0.9118 | 0.8785 | 13.41 | 142.5 | 6.958 | 0.9901 |
| | i-ResNet | 0.0502 | 0.6422 | 0.9149 | 0.8922 | 1.876 | 19.28 | 10.78 | **0.9925** |
| | i-ML-Enc(L-1) | **0.0407** | 0.5085 | **0.9986** | **0.9973** | **1.256** | **5.201** | **5.895** | 0.9580 |

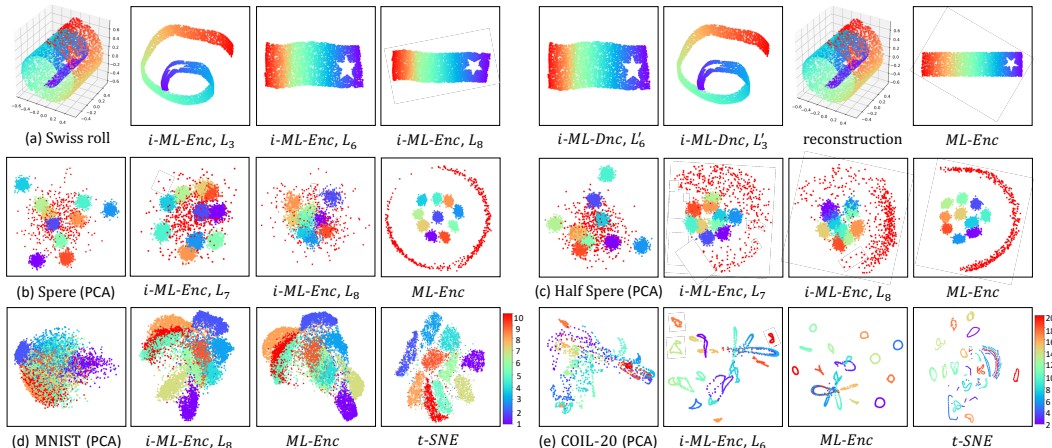

Figure 4: Visualization of invertible NLDR results of *i-ML-Enc* compared to ML-Enc and t-SNE. All the high-dimensional results are visualized by PCA and the target dimension $s' = 2$. (a) shows the NLDR and its inverse process of *i-ML-Enc* on the test set of Swiss roll in the case of $d = s = s'$. We show the cases of $s' < d \leq s$ and $s' = d \leq s$ by comparing (b)(c): (b) shows the failure case of reducing spheres $S^{100}$ sampled in $\mathbb{R}^{101}$ into 10-D, while (c) shows results of reducing half-spheres $S^{10}$ sampled in $\mathbb{R}^{101}$ into 10-D. The $L_7$ layers of *i-ML-Enc* show the same topology as the input data in both cases, but ML-Enc shows bad topological structures. (d) and (e) show results of two sparse cases on MNIST and COIL-20: The clustering effects of ML-Enc and t-SNE show the local geometric structure but dropping the relationship between sub-manifolds. With both of the geometric and topological structures, *i-ML-Enc* provides more reliable representations of the data manifold.

**Evalution metrics.** We evaluate an invertible NLDR algorithm from three aspects: (1) Invertible property. Reconstruction MSE (**RMSE**) and maximum norm error (**MNE**) measure the difference between the input data and reconstruction results by norm-based errors. (2) NLDR quality. Trustworthiness (**Trust**) and Continuity (**Cont**) (Kaski & Venna, 2006), latent MSE (**l-MSE**), Minimum ($K\mathbf{min}$) and Maximum ($K\mathbf{max}$) local Lipschitz constant (Li et al., 2020) are used to evaluate the quality of the low-dimensional representation. (3) Generalization ability of the representation. Mean accuracy (**Acc**) of linear classification on the representation measures models' generalization ability to downstream tasks. Their exact definitions and purpose are given in **Appendix A.1**.

**Conclusion.** Table 1 compares the *i-ML-Enc* with the related methods on MNIST, more results and detailed analysis on other datasets are given in **Appendix A.2**. The process of invertible NLDR of *i-ML-Enc* and comparing results of typical methods are visualized in Fig. 4. We can conclude: (1) *i-ML-Enc* achieves invertible NLDR in the first stage with great NLDR and generalization qualities. The representation in the $L - 1$-th layer of *i-ML-Enc* mostly outperforms all comparing methods for both invertible and NLDR metrics without losing information of the data manifold, while other methods drop geometric and topological information to some extent. (2) *i-ML-Enc* tries to keep more geometric and topological structure in the second stage in the case of $s' < d \leq s$. Though the representation of the $L$-th layer of *i-ML-Enc* achieves the second best in NLDR metrics, it shows high consistency with the $L - 1$-th layer in visualization results.

## 4.2 LATENT SPACE INTERPOLATION

Since the first stage of *i-ML-Enc* is nearly homeomorphism, we carry out linear interpolation experiments on the discrete data points in both the input space and the $(L - 1)$-th layer latent space to analyze the intrinsic continuous manifold, and verify the latent results by its inverse process. A good low-dimensional representation of the manifold should not only preserve the local properties, but also be flatter and denser than the high-dimensional input with lower curvature. Thus, we expect that the local linear interpolation results in the latent space should be more reliable than in the input space. The complexity of data manifolds increases from USPS(256), MNIST(256), MNIST(784), KMNIST(784) to FMNIST(784), which is analyzed in **Appendix A.3.1**.
**K-nearest neighbor interpolation.** We first verify the reliability of the low-dimensional repre-

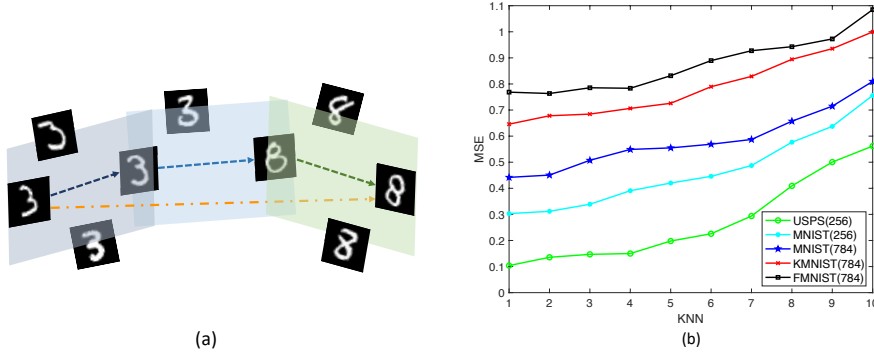

(a)

(b)

Figure 5: (a) shows the proposed geodesics interpolation on a manifold; (b) reports the MSE loss of 1 to 10 nearest neighbors interpolation results on interpolation datasets.

sentation in a small local system by kNN interpolation. Given a sample $x_i$, randomly select $x_j$ in $x_i$'s k-nearest neighborhood in the latent space to form a sample pair $(x_i, x_j)$. Perform linear interpolation of the latent representation of the pair and get reconstruction results for evaluation as: $\hat{x}_{i,j}^t = \psi^{-1}(t\psi(x_i) + (1-t)\psi(x_j))$, $t \in [0,1]$. The experiment is performed on *i-ML-Enc* with $L = 6$ and $K = 15$, training with 8000 samples for USPS and MNIST(256), 20000 sapmles for MNIST(784), KMNIST, FMNIST.

**Evaluation.** (1) Calculate the MSE loss between reconstruction results of the latent interpolation $\hat{x}_{i,j}^t$ and the input space result $x_{i,j}^t$ which is the corresponding interpolation results in the local neighborhood of the input space with $x_{i,j}^t = tx_i + (1-t)x_j$. Fig. 5 shows the results of $k = 1, 2, ..., 10$. (2) Visualize the typical results of the input space and the latent space for comparison, as shown in Fig. 6. More results and detailed analysis are given in **Appendix A.3.2**.

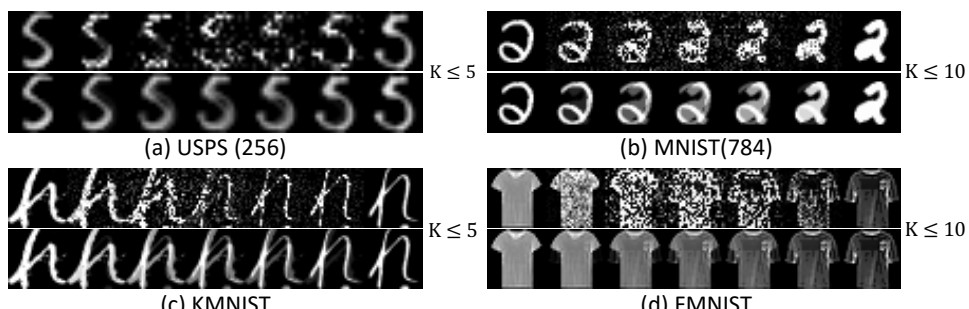

(a) USPS (256)    K ≤ 5

(b) MNIST(784)    K ≤ 10

(c) KMNIST    K ≤ 5

(d) FMNIST    K ≤ 10

Figure 6: The results of kNN interpolation in latent space. For each dataset, the upper row shows the latent result, while the lower shows the input result. The latent results show more noise but less overlapping and pseudo-contour than the input results.

**Geodesic interpolation.** Based on 4.2.1, we further employ a more reasonable method to generate the sampling points between two distant samples pairs in the latent space. Given a sample pair $(x_i, x_j)$ with $k \geq 45$ from different clusters, we select the three intermediate sample pairs $(x_i, x_{i_1})$, $(x_{i_1}, x_{i_2})$, $(x_{i_2}, x_j)$ with $k \leq 20$ along the geodesic path in latent space for piece-wise linear interpolation in both space. Visualization results are given in **Appendix A.3.2**.

**Conclusion.** Compared with results of the kNN and geodesic interpolation, we can conclude: (1) Because of the sparsity of the high-dimensional latent space, noises are inevitable on the latent results indicating the limitation of linear approximation. Empirically, the reliability of the latent interpolation decreases with the expansion of the local neighborhood on the same dataset. (2) We will get worse latent results in the following cases: on the similar manifolds, the sampling rate is lower or the input dimension is higher indicated by USPS(256), MNIST(256) and MNIST(784); with the same sampling rate and input dimension, the manifold is more complex indicated by MNIST(784), KMNIST to FMNIST. They indicate that the confidence of the tangent space estimated by local

neighborhood decreases on more complex manifolds with sparse sampling. (3) The interpolation between two samples in latent space is smoother than that in the input space, validating the flatness and density of the lower-dimensional representation learned by *i-ML-Enc*. Overall, we infer that the unreliable approximation of the local tangent space by the local neighborhood is the basic reason for the manifold learning fails in the real-world case, because the geometry should be preserved in the first place. To come up with this common situation, it is necessary to import other prior assumption or knowledge when the sampling rate of the data manifold is quite low, e.g. the Euclidean space assumption, semantic information of down-steam tasks.

### 4.3 ABLATION STUDY

**Analysis on loss terms.**    We perform an ablation study on MNIST, USPS, KMNIST, FMNIST and COIL-20 to evaluate the effects of the proposed network structure and loss terms in *i-ML-Enc* for invertible manifold learning. Based on ML-Enc, three proposed parts are added: the *extra head* (**Ex**), the orthogonal loss $\mathcal{L}_{orth}$ (**Orth**), the zero padding loss $\mathcal{L}_{pad}$ (**Pad**). Besides the previous 8 indicators, we introduce the rank of the output matrix of the layer $L-1$ as $r(Z^{L-1})$, to measure the sparsity of the high-dimensional representation. We conclude that the combination **Ex+Orth+Pad** is the best to achieve invertible NLDR of $s$-sparse by a series of equidimensional layers. The detailed analysis of experiment results are given in **Appendix A.4.1**.

**Orthogonality and sparsity.**    We further discuss the orthogonality of weight matrices and learned $s$-sparse representations in the first stage of *i-ML-Enc*. We find that the first $L-1$ layers of *i-ML-Enc* are nearly strict orthogonal mappings and the output from the $L-1$-th layer can be converted to $s$-dimensional representation without information loss. The detailed analysis are provided in **Appendix A.4.2**. Thus, we conclude that an invertible NLDR of data manifolds can be learned by *i-ML-Enc* in the *sparse coordinate transformation*.

## 5 CONCLUSION

A novel invertible NLDR process *inv-ML* and a neural network implementation *inv-ML-Enc* are proposed to tackle two problems of manifold-based DR in practical scenarios, i.e., the condition for information-lossless NLDR and the key issue of manifold learning. Firstly, the *sparse coordinate transformation* is learned to find a flatter and denser low-dimensional representation with preservation of geometry and topology of data manifolds. Secondly, we discuss the information loss with different target dimensions in *linear compression*. Experiment results of *i-ML-Enc* on seven datasets validate its invertibility. Further, the interpolation experiments reveal that finding a reliable tangent space by the local neighborhood on real-world datasets is the inherent defect of manifold based DR methods.

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

## A   APPENDIX

### A.1   DEFINITIONS OF PERFORMANCE METRICS

As for NLDR tasks, We adopt the performance metrics used in MLDL (Li et al., 2020) and TopoAE (Moor et al., 2020) to measure topology-based manifold learning, and add a new indicator to evaluate the generalization ability of the latent space. Essentially, the related indicators are defined based on comparisons of the local neighborhood of the input space and the latent representation. As for the invertible property, we adopted the norm-based reconstruction metrics, i.e. the $L_2$ and $L_\infty$ norm errors, which are also based on the inputs. The following notations are used in the definitions $d_{i,j}^{(l)}$ is the pairwise distance in space $Z^{(l)}$; $\mathcal{N}_{i,k}^{(l)}$ is the set of indices to the $k$-nearest neighbors ($k$-NN) of $z_i^{(l)}$ in latent space, and $\mathcal{N}_{i,k}$ is the set of indices to the $k$-NN of $x_i$ in input space; $r_{i,j}^{(l)}$ is the closeness rank of $z_j^{(l)}$ in the $k$-NN of $z_i^{(l)}$. The evaluation metrics are defined below:

(1) **RMSE** (invertible quality). This indicator is commonly used to measure reconstruction quality. Based on the input $x$ and the reconstruction output $\hat{x}$, the mean square error (MSE) of the $L_2$ norm is defined as:

$$RMSE = (\frac{1}{N^2} \sum_{i=1}^{N} (\boldsymbol{x}_i - \boldsymbol{z}_i)^2)^{\frac{1}{2}}.$$

(2) **MNE** (invertible quality). This indicator is designed to evaluate the bijective property of a $L$ layers neural network model. Specifically, taking each invertible unit in the network, calculate the $L_\infty$ norm error of the input and reconstruction output of each corresponding layer, and choose the maximum value among all units. If a model is bijective, this indicator can reflect the stability of the model:

$$MNE = \max_{1 \leq l \leq L-1} \|\boldsymbol{z}_l - \hat{\boldsymbol{z}}_l\|_\infty, \; l = 1, 2, ...L.$$

(3) **Trust** (embedding quality). This indicator measures how well neighbors are preserved between the two spaces. The $k$ nearest neighbors of a point are preserved when going from the input space $X$ to space $Z^{(l)}$:

$$Trust = \frac{1}{k_2 - k_1 + 1} \sum_{k=k_1}^{k_2} \left\{ 1 - \frac{2}{Mk(2M - 3k - 1)} \sum_{i=1}^{M} \sum_{j \in \mathcal{N}_{i,k}^{(l)}, j \notin \mathcal{N}_{i,k}} (r_{i,j}^{(l)} - k) \right\}$$

where $k_1$ and $k_2$ are the bounds of the number of nearest neighbors, so averaged for different $k$-NN numbers.

(4) **Cont** (embedding quality). This indicator is asymmetric to **Trust**. It checks to what extent neighbors are preserved from the latent space $Z^{(l)}$ to the input space $X$:

$$Cont = \frac{1}{k_2 - k_1 + 1} \sum_{k=k_1}^{k_2} \left\{ 1 - \frac{2}{Mk(2M - 3k - 1)} \sum_{i=1}^{M} \sum_{j \in \mathcal{N}_{i,k}, j \notin \mathcal{N}_{i,k}^{(l)}} (r_{i,j}^{(l)} - k) \right\}$$

(5) $K\mathbf{min}$ and $K\mathbf{max}$ (embedding quality). Those indicators are the minimum and maximum of the local bi-Lipschitz constant for the homeomorphism between input space and the $l$-th layer, with respect to the given neighborhood system:

$$K_{\min} = \min_{1 \leq i \leq M} \max_{j \in \mathcal{N}_{i,k}^{(l)}} K_{i,j}, \; K_{\max} = \max_{1 \leq i \leq M} \max_{j \in \mathcal{N}_{i,k}^{(l)}} K_{i,j},$$

where $k$ is that for $k$-NN used in defining $N_i$ and

$$K_{i,j} = \max \left\{ \frac{d_{i,j}^{(l)}}{d_{i,j}^{(l')}}, \frac{d_{i,j}^{(l')}}{d_{i,j}^{(l)}} \right\}.$$

(6) $l$-**MSE** (embedding quality). This indicator is to evaluate the distance disturbance between the input space and latent space with $L_2$ norm-based error.

$$lMSE = (\frac{1}{N^2} \sum_{i=1}^{N} \sum_{j=1}^{N} \|d_X(\boldsymbol{x}_i, \boldsymbol{x}_j) - d_Z(h(\boldsymbol{x}_i), h(\boldsymbol{x}_j))\|)^{\frac{1}{2}}.$$

(7) **ACC** (generalization ability). In general, a good representation should have a good generation ability to downstream tasks. To measure this ability, logistic regression (Pedregosa et al., 2011) is performed after the learned latent representation. We report the mean accuracy on the test set for 10-fold cross-validation.

## A.2 METHOD COMPARISON

**Configurations of datasets.** The NLDR performance and its inverse process are verified on both synthetic and real-world datasets. As shown in Table 2, we list the **type** of the dataset, the **class** number of clusters, the **input** dimension $m$, the **target** dimension $s'$, the **intrinsic** dimension $d$ which is only an approximation for the real-world dataset, the number of **train** and **test** samples, and the **logistic** classification performance on the raw input space. Among them, Swiss roll serves as an ideal example of information-lossless NLDR; Spheres, whose target dimension $s'$ is lower than the intrinsic dimension $s$, serves as an excessive case of NLDR compared to Half-spheres; four image datasets with increasing difficulties are used to analyze complex situations in real-world scenarios. Additionally, the lower bound and upper bound of the intrinsic dimension of real-world datasets are approximated by (Hein & Audibert, 2005) and AE-based INN (Nguyen et al., 2019). Specifically, the upper bound can be found by the grid search of different bottlenecks of the INN, and we report the bottleneck size of each dataset when the reconstruction MSE loss is almost unchanged.

Table 2: Brief introduction to the configuration of datasets for method comparison.

| Dataset | Type | Class | Input $m$ | Target $s'$ | intrinsic $d$ | Train | Test | Logistic |
|---------|------|-------|-----------|-------------|---------------|-------|------|----------|
| Swiss roll | synthetic | - | 3 | 2 | 2 | 800 | 8000 | - |
| Spheres | synthetic | - | 101 | 10 | 101 | 5500 | 5500 | - |
| Half-spheres | synthetic | - | 101 | 10 | 10 | 5500 | 5500 | - |
| USPS | real-world | 10 | 256 | 10 | 10 to 80 | 4649 | 4649 | 0.9381 |
| MNIST | real-world | 10 | 784 | 10 | 10 to 100 | 20000 | 10000 | 0.8943 |
| FMNIST | real-world | 10 | 784 | 10 | 20 to 140 | 20000 | 10000 | 0.7984 |
| COIL-20 | real-world | 20 | 4096 | 20 | 20 to 260 | 1440 | 1440 | 0.9974 |

**Hyperparameter values.** Basically, *i-ML-Enc* is trained with Adam optimizer (Kingma & Ba, 2015) and learning rate $lr = 0.001$ for 8000 epochs. We set the layer number $L = 8$ for most datasets but $L = 6$ for COIL-20. The bound in push-away loss is set $B = 3$ in most datasets but removed in Spheres and Half-spheres. We set the hyperparameter based on two intuitions: (1) the implementation of *sparse coordinate transformation* should achieve DR on the premise of maintaining homeomorphism; (2) NLDR should be achieved gradually from the first to $(L-1)$-th layer because NLDR is impossible to achieve by a single nonlinear layer. Based on (1), we decrease the *extra heads* weights $\gamma$ linearly for epochs from 2000 to 8000, while linearly increase the orthogonal loss weights $\alpha$ for epochs from 500 to 2000. Based on (2), we approximate the DR trend by exponential series. For the *extra heads*, the target dimension decrease exponentially from $m$ to $s'$ for the 2-th to $(L-1)$-th layer, and the push-away loss weights $\mu$ increase linearly. Similarly, the padding weight $\beta$ should increase linearly. Because the intrinsic dimension is different from each real-world dataset, we adjust the prior hyperparameters according to the approximated intrinsic dimension.

**Results on toy datasets.** The Table 3 compares the *i-ML-Enc* with other methods in 9 performance metrics on Swiss roll and Half-spheres datasets in the case of $s = s'$. Eight methods for manifold learning: Isomap (Tenenbaum et al., 2000), t-SNE (Maaten & Hinton, 2008), RR (McQueen et al., 2016), and ML-Enc (Li et al., 2020) are compared for NLDR; four AE-based methods AE (Hinton & Salakhutdinov, 2006), VAE (Kingma & Welling, 2014), TopoAE (Moor et al., 2020), and ML-AE (Li et al., 2020) are compared for reconstructible manifold learning. We report the $L$-th layer of *i-ML-Enc* (the first stage) for the NLDR quality and the $(L-1)$-th layer (the second stage) for the

Table 3: Comparison in embedding and invertible quality on Swiss roll and Half-spheres datasets. *I-ML-Enc* achieves invertible NLDR in the first stage and top three embedding performance in the second stage when $s' = d = s$.

| Dataset | Algorithm | RMSE | MNE | Trust | Cont | $K$min | $K$max | $l$-MSE |
|---|---|---|---|---|---|---|---|---|
| Swiss Roll | Isomap | - | - | 0.9834 | 0.9812 | 1.213 | 43.55 | 0.0756 |
| | t-SNE | - | - | 0.9987 | 0.9843 | 10.96 | 1097 | 3.407 |
| | RR | - | - | 0.9286 | 0.9847 | 4.375 | 187.7 | 0.0453 |
| | ML-Enc | - | - | **0.9999** | 0.9985 | **1.000** | **2.141** | **0.0039** |
| | AE | 0.3976 | 10.55 | 0.8724 | 0.8333 | 1.687 | 4230 | 0.0308 |
| | VAE | 0.7944 | 13.97 | 0.5064 | 0.6486 | 1.51 | 4809 | 0.0397 |
| | TopoAE | 0.5601 | 11.61 | 0.9198 | 0.9881 | 1.194 | 220.6 | 0.1165 |
| | ML-AE | 0.0208 | 8.134 | 0.9998 | 0.9847 | 1.005 | 2.462 | 0.0051 |
| | i-ML-Enc (L) | **0.0048** | **0.0649** | 0.9996 | **0.9986** | 1.004 | 2.471 | 0.0043 |
| Half-spheres | Isomap | - | - | 0.8701 | 0.9172 | 1.845 | 199.3 | 0.4046 |
| | t-SNE | - | - | **0.8908** | 0.9278 | 25.33 | 790.9 | 2.6665 |
| | RR | - | - | 0.8643 | 0.8516 | 3.047 | 201.2 | 0.4789 |
| | ML-Enc | - | - | 0.8837 | 0.9305 | 1.029 | 46.35 | **0.0207** |
| | AE | 0.7359 | 11.54 | 0.6886 | 0.7069 | 1.763 | 4112 | 0.0937 |
| | VAE | 0.8474 | 14.97 | 0.5398 | 0.6197 | 2.361 | 4682 | 0.1205 |
| | TopoAE | 0.9174 | 13.68 | 0.8574 | 0.8226 | 1.375 | 154.8 | 0.4342 |
| | ML-AE | 0.6339 | 9.492 | 0.8819 | 0.9293 | **1.025** | 43.17 | 0.0218 |
| | i-ML-Enc (L) | **0.1095** | **0.7985** | 0.8892 | **0.9295** | 1.491 | **41.25** | 0.0463 |

Table 4: Comparison in embedding and invertible quality on USPS, FMNIST, and COIL-20 datasets. ML-Enc shows comparable performance for embedding metrics. Based on ML-Enc, *i-ML-Enc* achieves invertible NLDR in the first stage while maintaining a good generalization ability. It also achieves the top embedding performance for the most NLDR metrics in the second stage when $s' < d \leq s$.

| Dataset | Algorithm | RMSE | MNE | Trust | Cont | $K$min | $K$max | $l$-MSE | Acc |
|---|---|---|---|---|---|---|---|---|---|
| USPS | t-SNE | - | - | 0.9831 | 0.9889 | 3.238 | 194.8 | 35.53 | 0.9522 |
| | ML-Enc | - | - | 0.9874 | 0.9897 | 1.562 | **52.14** | **14.88** | 0.9534 |
| | AE | 0.6201 | 29.09 | 0.9845 | 0.974 | 4.728 | 87.41 | 17.41 | 0.8952 |
| | TopoAE | 0.647 | 30.19 | 0.9830 | 0.9852 | 3.598 | 126.2 | 19.98 | 0.8876 |
| | ML-AE | 0.4912 | 11.84 | 0.9879 | **0.9905** | 1.529 | 55.32 | 15.05 | **0.9576** |
| | i-ML-Enc (L) | **0.0253** | **0.3058** | **0.9886** | 0.9861 | **1.487** | 60.79 | 15.16 | 0.9435 |
| | INN | 0.0535 | 0.5239 | 0.9872 | 0.9843 | 1.795 | 26.38 | 9.581 | 0.9305 |
| | i-RevNet | 0.0337 | 0.3471 | 0.9187 | 0.9096 | 11.25 | 183.2 | 6.209 | 0.9945 |
| | i-ResNet | 0.0437 | 0.5789 | 0.9205 | 0.9122 | 1.635 | 18.375 | 9.875 | **0.9974** |
| | i-ML-Enc(L-1) | **0.0253** | **0.3058** | **0.9934** | **0.9927** | **1.165** | **4.974** | **5.461** | 0.9876 |
| FMNIST | t-SNE | - | - | 0.9896 | 0.9863 | 3.247 | 108.3 | 48.07 | 0.7249 |
| | ML-Enc | - | - | 0.9903 | 0.9896 | 1.358 | 89.65 | 25.18 | 0.7629 |
| | AE | 0.2078 | 27.45 | 0.9744 | 0.9689 | 6.728 | 102.1 | 21.98 | 0.7495 |
| | TopoAE | 0.2236 | 31.01 | 0.9658 | 0.9813 | 6.982 | 115.4 | 23.53 | 0.7503 |
| | ML-AE | 0.4912 | 18.84 | 0.9912 | **0.9917** | 1.738 | 101.7 | 25.89 | **0.7665** |
| | i-ML-Enc (L) | **0.0461** | **0.3567** | 0.9923 | 0.9905 | **1.295** | **83.63** | **20.13** | 0.7644 |
| | INN | 0.0627 | 0.6819 | 0.9832 | 0.9744 | 1.364 | 21.36 | 9.258 | 0.8471 |
| | i-RevNet | 0.0475 | **0.3519** | 0.9157 | 0.8967 | 21.58 | 204.8 | 6.517 | 0.9386 |
| | i-ResNet | 0.0582 | 0.6719 | 0.9242 | 0.9058 | 1.953 | 22.75 | 9.687 | **0.9477** |
| | i-ML-Enc(L-1) | **0.0461** | 0.3567 | **0.9935** | **0.9959** | 1.356 | **6.704** | **6.017** | 0.8538 |
| Coil-20 | t-SNE | - | - | 0.9911 | **0.9954** | 5.794 | 101.2 | 17.22 | 0.9039 |
| | ML-Enc | - | - | 0.9920 | 0.9889 | 1.502 | 70.79 | **9.961** | **0.9564** |
| | AE | 0.3507 | 24.09 | 0.9745 | 0.9413 | 4.524 | 85.09 | 11.45 | 0.8958 |
| | TopoAE | 0.4712 | 26.66 | 0.9768 | 0.9625 | 5.272 | 98.33 | 27.19 | 0.9043 |
| | ML-AE | 0.1220 | 16.86 | 0.9914 | 0.9885 | **1.489** | **68.63** | 10.34 | 0.9548 |
| | i-ML-Enc (L) | **0.0312** | **1.026** | **0.9921** | 0.9871 | 1.695 | 71.86 | 11.13 | 0.9386 |
| | INN | 0.0758 | 0.8075 | 0.9791 | 0.9681 | 2.033 | 79.25 | 8.595 | 0.9936 |
| | i-RevNet | 0.0508 | 0.7544 | 0.9316 | 0.9278 | 11.34 | 147.2 | 9.803 | **1.000** |
| | i-ResNet | 0.0544 | **0.7391** | 0.9258 | 0.9136 | 1.821 | 13.56 | 10.41 | **1.000** |
| | i-ML-Enc(L-1) | **0.0312** | 0.9263 | **0.9940** | **0.9937** | **1.297** | **4.439** | **7.539** | **1.000** |

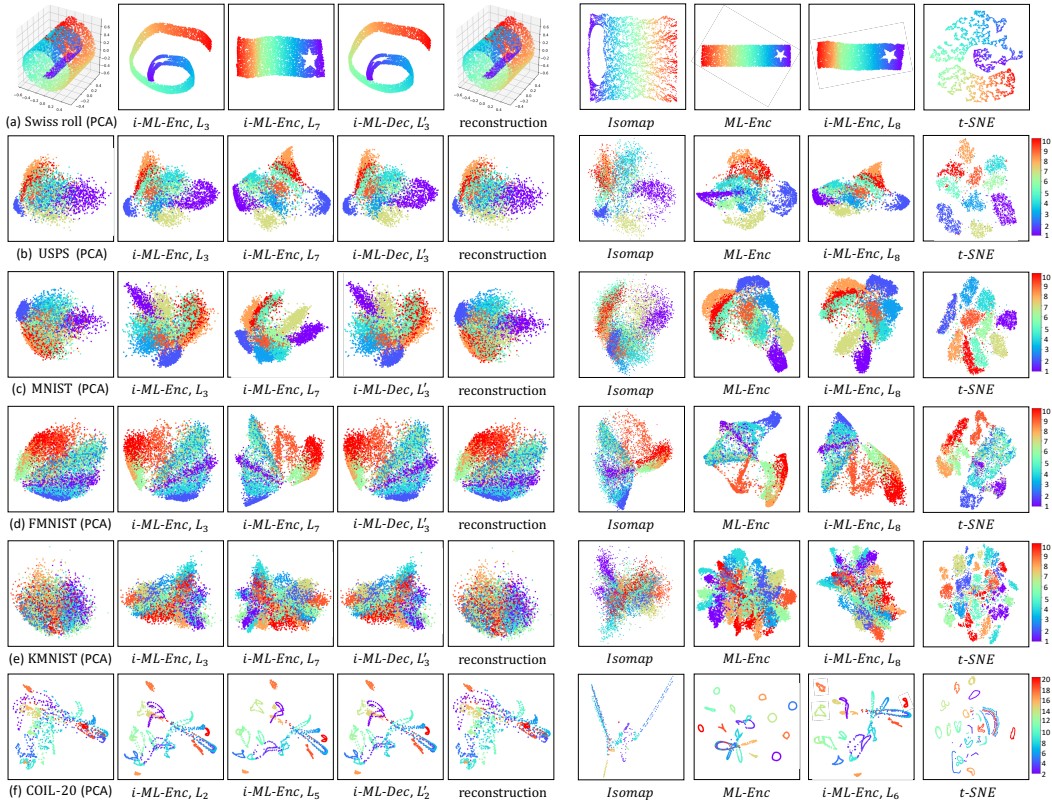

Figure 7: Visualization of invertible NLDR results of *i-ML-Enc* with comparison to Isomap, ML-Enc, and t-SNE on Swiss roll and five real-world datasets. The target dimension $s' = 2$ for all datasets, and the high-dimensional latent space are visualized by PCA. For each row, the left five cells show the NLDR and reconstruction process in the first stage of *i-ML-Enc*, and the right four cells show 2D results for comparison. ML-Enc and t-SNE show great clustering effects but drop topological information. Compared to the classical DR method Isomap (preserving the global geodesic distance) and t-SNE (preserving the local geometry), the representations learned by *i-ML-Enc* preserves the relationship between clusters and the local geometry within clusters.

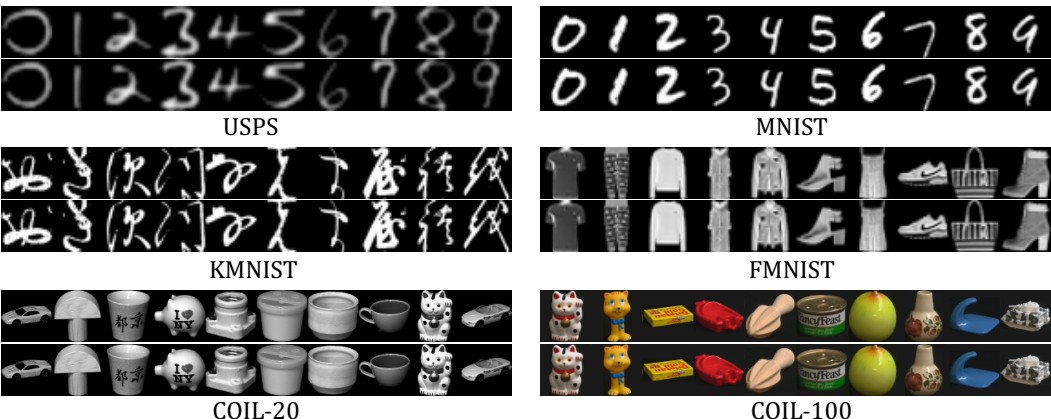

Figure 8: Visualization of reconstruction results of *i-ML-Enc* on six image datasets. For each cell, the upper row shows results of *i-ML-Enc* while the lower row shows the raw input images. We randomly selected 10 images from different classes to demonstrate the bijective property of *i-ML-Enc*.

invertible NLDR ability. ML-Enc performs best in Trust, $K$min, $K$max, and $l$-MSE on Swiss roll which shows its great embedding abilities. Based on ML-Enc, *i-ML-Enc* achieves great embedding results in the second stage on Half-spheres, which shows its advantages of preserving topological and geometric structures in the high-dimensional case. And *i-ML-Enc* outperforms other methods in its invertible NLDR property of the first stage.

**Results on real-world datasets.** The Table 4 compares the *i-ML-Enc* with other methods in 9 performance metrics on USPS, FMNIST and COIL-20 datasets in the case of $s > s'$. Six methods for manifold learning: Isomap, t-SNE, and ML-Enc are compared for NLDR; three AE-based methods AE, ML-AE, and TopoAE are compared for reconstructible manifold learning. Three methods for inverse models: INN (Nguyen et al., 2019), i-RevNet (Jacobsen et al., 2018), and i-ResNet (Behrmann et al., 2019) are compared for bijective property. The visualization of NLDR and its inverse process of *i-ML-Enc* are shown in Fig. 7, together with the NLDR results of Isomap, t-SNE and, ML-Enc. The target dimension for visualization is $s' = 2$ and the high-dimensional latent space are visualized by PCA. Compared with NLDR algorithms, the representation of the $L$-th layer of *i-ML-Enc* nearly achieves the best NLDR metrics on FMNIST, and ranks second place on USPS and third place on COIL-20. The drop of performance between the $(L-1)$-th and $L$-th layers of *i-ML-Enc* are caused by the sub-optimal linear transformation layer, since the representation of its first stage are quite reliable. Compared with other inverse models, *i-ML-Enc* outperforms in all the NLDR metrics and inverse metrics in the first stage, which indicates that a great low-dimensional representation of data manifolds can be learned by a series of equidimensional layers. However, *i-ML-Enc* shows larger NME on FMNIST and COIL-20 compared with inverse models, which indicates that *i-ML-Enc* is more unstable dealing with complex datasets in the first stage. Besides, we visualize the reconstruction samples of six image datasets including COIL-100 (Nene et al., 1996a) to show the inverse quality of *i-ML-Enc* in Fig. 8.

## A.3 LATENT SPACE INTERPOLATION

### A.3.1 DATASETS COMPARISON

Here is a brief introduction to four interpolation data sets. We analyze the difficulty of dataset roughly according to **dimension**, **sample size**, **image entropy**, **texture**, and the performance of **classification tasks**: (1) Sampling ratio. The input dimension and sample number reflect the sampling ratio. Generally, the sample number has an exponential relationship with the input dimension in the case of sufficient sampling. Thus, the sampling ratio of USPS is higher than others. (2) Image entropy. The Shannon entropy of the histogram measures the information content of the image, and it reaches the maximum when the density estimated by the histogram is an uniform distribution. We report the mean entropy of each dataset. We conclude that USPS has richer grayscale than MNIST(256), while the information content of MNIST(784), KMNIST, and FMNIST shows an increasing trend. (3) Texture. The standard deviation (std) of the histogram reflects the texture information in the image, and we report the mean std of each dataset. Combined with the evaluation of human eyes, the texture features become rougher and rougher from USPS, MNIST to KMNIST, while FMNIST contains complex and regular texture. (4) Classification tasks. We report the mean accuracy of 10-fold cross-validation using kNN and logistic classifier (Pedregosa et al., 2011) for each data set. The credibility of the neighborhood system decreases gradually from USPS, MNIST, KMNIST to FMNIST. Combined with the visualization results of each dataset in Fig. 7, it obvious that KMNIST has the worst linear separability. Above all, we can roughly give the order of the difficulty of manifold learning on each data set: **USPS<MNIST(256)<MNIST(784)<KMNIST<FMNIST**.

Table 5: Comparison of manifold learning difficulties of interpolation datasets

| Dataset | Class | Train set | Dimension | Entropy | Texture | KNN | Logistic |
|---------|-------|-----------|-----------|---------|---------|------|----------|
| USPS | 10 | 9298 | 256 | 5.479 | 0.5097 | 0.9589 | 0.9381 |
| MNIST(256) | 10 | 9298 | 256 | 1.879 | 10.51 | 0.9493 | 0.9099 |
| MNIST(784) | 10 | 20000 | 784 | 1.598 | 39.75 | 0.9515 | 0.8943 |
| KMNIST | 10 | 20000 | 784 | 2.911 | 33.01 | 0.9141 | 0.6471 |
| FMNIST | 10 | 20000 | 784 | 4.115 | 24.75 | 0.8133 | 0.7984 |

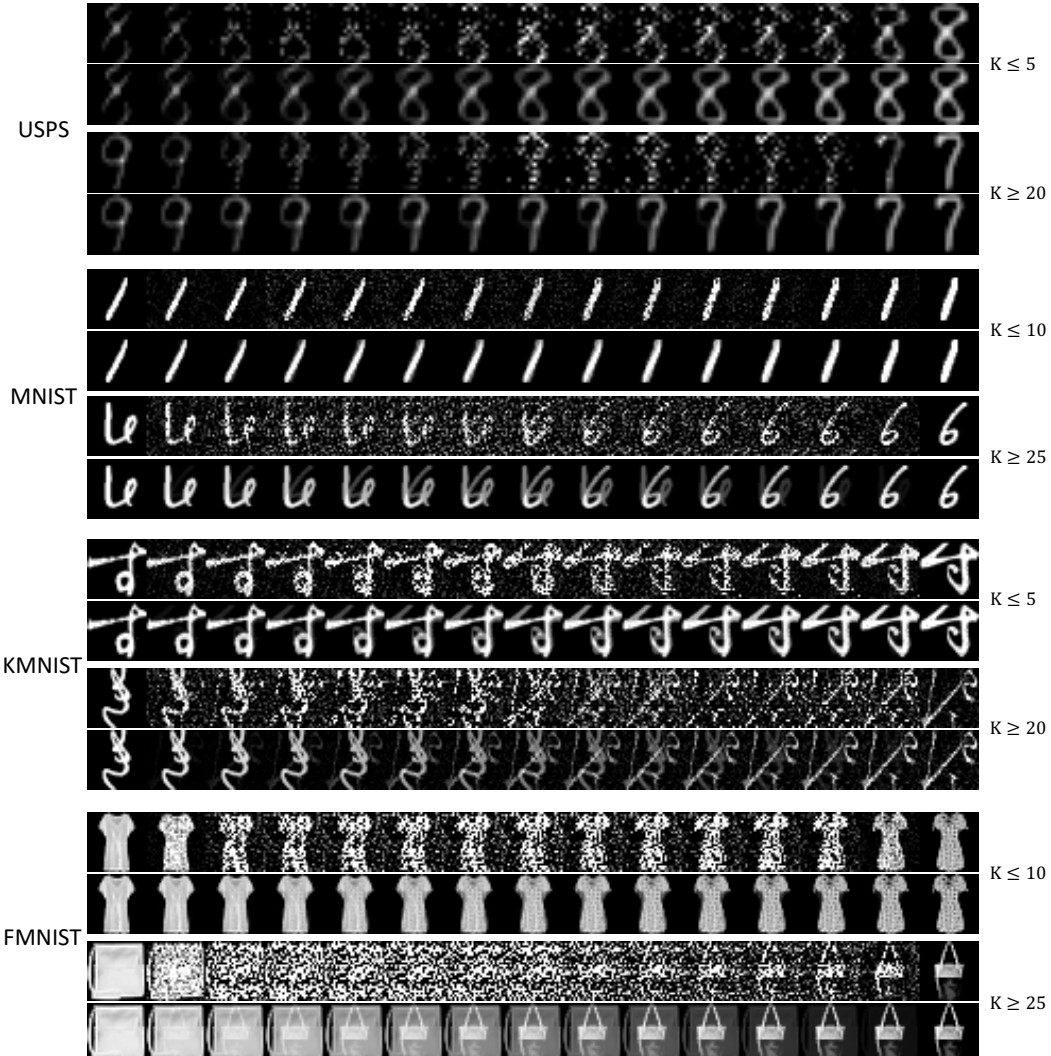

Figure 9: Visualization of kNN interpolation results of *i-ML-Enc* on image datasets with $k \leq 10$ and $k \geq 20$. For each row, the upper part shows results of *i-ML-Enc* while the lower part shows the raw input images. Both the input and latent results transform smoothly when $k$ is small, while the latent results show more noise but less overlapping and pseudo-contour than the input results when $k$ is large. The latent interpolation results show more noise and less smoothness when the data manifold becomes more complex.

### A.3.2 MORE INTERPOLATION RESULTS

**kNN interpolation.** We verify the reliability of the low-dimensional representation by kNN interpolation. Comparing the results of different values of $k$, as shown in Fig. 9, we conclude that: (1) Because the high-dimensional latent space is still quite sparse, there is some noise caused by linear approximation on the latent results. The MSE loss and noises of the latent results are increasing with the expansion of the local neighborhood on the same dataset, reflecting the reliability of the local neighborhood system. (2) In terms of the same sampling rate, the MSE loss and noises of the latent results grow from MNIST(784), KMNIST to FMNIST, which indicates that the confidence of the local homeomorphism property of the latent space decreases gradually on more difficult manifolds. (3) In terms of the similar data manifolds, USPS(256) and MNIST(256) show better latent interpolation results than MNIST(784), which demonstrates that it is harder to preserve the geometric properties on higher input dimension. (4) Though the latent results import some noise, the input results have unnatural transformations such as pseudo-contour and overlapping. Thus, the latent

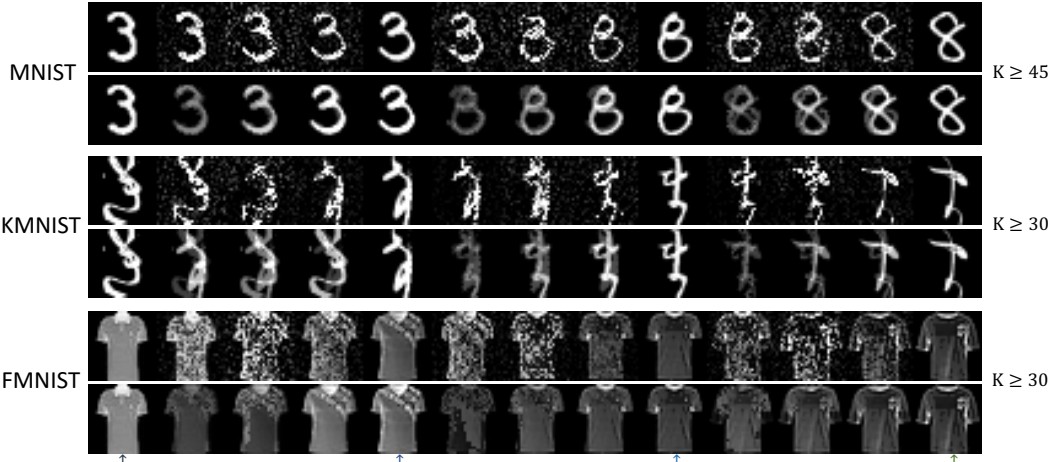

Figure 10: The interpolation results of the geodesic interpolation in the latent space. For each dataset, the upper row shows the latent result, while the lower shows the input result. The samples 1, 5, 9, 13 pointed by the arrow are the original samples.

space results are more smooth than the input space, which validates that the latent space learned by *i-ML-Enc* is flatter and denser than the input space. In a nutshell, we infer that the difficulty of preserving the geometric properties based on approximation of the local tangent space by the local neighborhood is the key reason for the manifold learning fails in the real-world case.

**Geodesic interpolation.** We further perform the latent interpolation along the geodesic path between sample pairs when $k$ is large to generate reliable intermediate samples. It might reflect the topological structure of data manifolds when two samples in a sample pair are in different clusters. Compared with results of MNIST, KMNIST, and FMNIST, as shown in Fig. 10, we can conclude: (1) The latent results are more reliable than those in the input space which can generate the synthetic samples between two different clusters. (2) Compared with MNIST, KMNIST, and FMNIST, the latent results of more complex datasets are more ambiguous and noisy, which indicates that it is more difficult to find a low-dimensional representation of more complex data manifolds with all geometric structure preserved.

## A.4 ABLATION STUDY

### A.4.1 ANALYSIS OF THE LOSS TERMS

We further conduct ablation study of the *extra head* (**+Ex**), the orthogonal loss $\mathcal{L}_{orth}$ (**+Orth**), and the zero padding loss $\mathcal{L}_{pad}$ (**+Pad**) on MNIST, USPS, KMNIST, FMNIST and COIL-20. The Table 6 reports ablation results in the 8 indicators and the $r(Z^{L-1})$. We analyze and conclude: (1) The combination of **Ex** and **Orth** nearly achieve the best inverse and DR performance on MNIST, USPS, FMNIST, and COIL-20, which indicates that it is the basic factor for invertible NLDR in the first $L-1$ layers. (2) When only use **Orth**, the NLDR in the first $L-1$ layer of the network will degenerate into the identity mapping, and DR is achieved with the linear project on layer $L$. (3) Combined with all three items **Ex**, **Orth** and **Pad**, *i-ML-Enc* obtains a sparse coordinate representation, but achieves little worse embedding quality on USPS and COIL-20 than using **Ex** and **Orth**. (4) Besides the proposed loss items, ML-AE overperforms the other combinations in the **Acc** metric indicating the reconstruction loss helps improve the generation ability of ML-Enc. Above all, the **Ex+Orth+Pad** combination, i.e. *i-ML-Enc*, can achieve the proposed invertible NLDR.

### A.4.2 ORTHOGONALITY AND SPARSITY

**Orthogonal analysis.** We first analyze the orthogonality of weight matrices in the first stage by evaluating the orthogonal loss $||W_l^T W_l - I||$. Using the same experimental settings as Sec 4.1, the maximum of non-diagonal elements and the minimum of diagonal elements of each layer are calculated in the first stage of *i-ML-Enc* on different datasets. We find that the margin of the maximum

value and the minimum value is at least $4$ orders of magnitude apart, as shown in Fig. 11. We can conclude that the first $L-1$ layers in *i-ML-Enc* are close to strict orthogonal mappings.

Table 6: Ablation study of the proposed loss terms in *i-ML-Enc* on five image datasets.

| Dataset | Algorithm | RMSE | MNE | Trust | Cont | $K$min | $K$max | Acc | $r(Z^{L-1})$ |
|---|---|---|---|---|---|---|---|---|---|
| MNIST | ML-AE | 0.4012 | 16.84 | 0.9893 | 0.9926 | 1.704 | 57.48 | **0.9340** | 15 |
| | ML-Enc | - | - | 0.9862 | **0.9927** | 1.761 | 58.91 | 0.9326 | 14 |
| | +Ex | - | - | 0.9891 | 0.9812 | 2.745 | 78.88 | 0.9316 | **12** |
| | +Ex+Orth | 0.0341 | 0.4255 | 0.9874 | **0.9927** | 1.817 | 59.97 | 0.9298 | 361 |
| | +Ex+Orth+Pad | 0.0457 | 0.5085 | **0.9906** | 0.9912 | 2.033 | 60.14 | 0.9316 | 125 |
| | +Orth | **0.0056** | **0.1275** | 0.9652 | 0.9578 | **1.597** | **53.21** | 0.8807 | 716 |
| USPS | ML-AE | 0.4912 | 11.84 | 0.9879 | **0.9905** | 1.529 | 55.32 | **0.9576** | 16 |
| | ML-Enc | - | - | 0.9874 | 0.9897 | 1.562 | **52.14** | 0.9534 | 14 |
| | +Ex | - | - | 0.9849 | 0.9836 | 2.525 | 78.88 | 0.9413 | **11** |
| | +Ex+Orth | 0.0395 | 0.2511 | **0.9895** | 0.9875 | 1.366 | 58.83 | 0.9376 | 192 |
| | +Ex+Orth+Pad | 0.0253 | 0.3058 | 0.9886 | 0.9861 | 1.538 | 60.79 | 0.9456 | 116 |
| | +Orth | **0.0109** | **0.2043** | 0.9702 | 0.9654 | **1.328** | 66.25 | 0.8961 | 243 |
| KMNIST | ML-AE | 0.4912 | 18.84 | 0.9781 | **0.9912** | 2.478 | 80.66 | 0.7639 | 19 |
| | ML-Enc | - | - | 0.9738 | 0.9883 | 2.253 | 103.4 | **0.7719** | **18** |
| | +Ex | - | - | 0.9786 | 0.9801 | 5.826 | 255.1 | 0.7624 | **18** |
| | +Ex+Orth | 0.0463 | 0.4661 | 0.9805 | 0.9872 | 2.396 | 70.89 | 0.6325 | 406 |
| | +Ex+Orth+Pad | 0.0844 | 0.4589 | **0.9875** | 0.9894 | 2.697 | 78.19 | 0.7513 | 198 |
| | +Orth | **0.0223** | **0.1962** | 0.9621 | 0.9593 | **1.991** | **60.51** | 0.5922 | 732 |
| FMNIST | ML-AE | 0.4912 | 18.84 | 0.9912 | **0.9917** | 1.738 | 101.7 | **0.7665** | 19 |
| | ML-Enc | - | - | 0.9903 | 0.9896 | 1.358 | 89.65 | 0.7629 | 18 |
| | +Ex | - | - | 0.9886 | 0.9726 | 5.826 | 279.4 | 0.7624 | **16** |
| | +Ex+Orth | 0.0337 | 0.3194 | 0.9895 | 0.9840 | 1.879 | 98.66 | 0.7613 | 393 |
| | +Ex+Orth+Pad | 0.0461 | 0.3567 | **0.9923** | 0.9905 | **1.298** | **83.63** | 0.7644 | 182 |
| | +Orth | **0.0152** | **0.2975** | 0.9701 | 0.9593 | 2.073 | 89.03 | 0.5934 | 743 |
| COIL-20 | ML-AE | 0.1220 | 16.87 | 0.9914 | 0.9885 | 1.489 | 74.79 | **0.9564** | **44** |
| | ML-Enc | - | - | 0.9920 | **0.9889** | 1.502 | 70.79 | **0.9564** | 46 |
| | +Ex+Orth | **0.0049** | **0.093** | **0.9927** | 0.9852 | **1.378** | **66.39** | 0.9427 | 1190 |
| | +Ex+Orth+Pad | 0.0171 | 1.026 | 0.9921 | 0.9871 | 1.695 | 71.86 | 0.9386 | 746 |

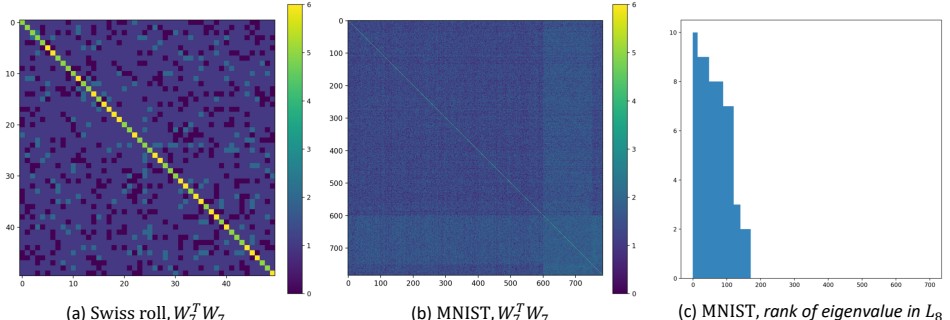

(a) Swiss roll, $W_7^T W_7$  (b) MNIST, $W_7^T W_7$  (c) MNIST, *rank of eigenvalue in* $L_8$

Figure 11: Visualization analysis of the representation from the $L-1$-th layer of *i-ML-Enc*. (a) and (b) show the margin of the non-diagonal elements and the diagonal elements with $W_7^T W_7$ trained on Swiss roll (50x50) and MNIST (784x784). These elements are divided into 7 orders of magnitude after min-max normalization, indicating the large margin of the diagonal and non-diagonal elements. (c) shows the rank of eigenvalues of the subspace decomposed (by SVD) from the output of the 8-th layer of *i-ML-Enc* on the test set of MNIST. The number of the main eigenvalues of the output is 125 which is equal to its matrix rank.

**Discussion on the $s$-sparse representation.** We first provide a possible way to decompose the learned low-dimensional representation in the $L-1$-th layer of *i-ML-Enc*, i.e. decomposing the output matrix by PCA to construct a linear subspace. Taking the 8-th layer of *i-ML-Enc* on the test set of MNIST which is 784-D as an example, we can construct a 171-D orthogonal base vectors in the linear

subspace from the data matrix after dimension reduction and reconstruct to the original space (784-D) without losing information by PCA (Pedregosa et al., 2011) and its inverse transform, as shown in Fig. 11. Compared to the matrix rank 125-D in the 784-D space, the extra 46-D in the subspace can be regarded as the machine error in the process of performing PCA because of the large margin between the first 125 eigenvalues and the rest. We notice that the $s$-sparse achieved by the first stage of *i-ML-Enc* is higher than the approximate intrinsic dimension $d$ on each dataset, e.g. 116-sparse on USPS and 125-sparse on MNIST. We found the following reasons: (1) Because the data manifolds are usually quite complex but sampling sparsely, the lowest isometric embedding dimension are between $d$ to $2d$ according to Nash Embedding Theorem and the hyper-plane hypothesis. The $s$ obtained by *i-ML-Enc* on each dataset is nearly in the interval of $[d, 2d]$, which is not the true intrinsic dimension of the manifolds. (2) The proposed *i-ML-Enc* is not optimized enough which serves as a simple network implementation of inv-ML. We need to design a better implementation model if we want to approach the lower embedding dimension with the preservation of both geometry and topology.

