# OpenReview forum: "Invertible Manifold Learning for Dimension Reduction"
_ICLR.cc/2021/Conference — Reject_

### Official Review · AnonReviewer1 · 2020-10-16
**Really invertible?**

**Rating:** 4
**Confidence:** 4

**Review:**

The paper presents a nonlinear dimensionality reduction (NLDR) method which is claimed to be invertible. The method inserts an existing idea (LIS) to a two-stage neural network implementation. The proposed method is tested with several known benchmark datasets as well as three synthetic datasets.

The paper currently is not enough for acceptance.

First of all, the problem in attack is unclear. By title, abstract and introduction, it reads that the target is a lossless NLDR method such that the data objects can be perfectly reconstructed from the subspace. However, this is not true in the experiment part. Clearly the loss is not zero after reconstruction. On the other hand, without perfect reconstruction, the proposed method is just another approximation and I don't see major contribution. So I believe the term "invertibility" is different from common understanding and needs more careful definition.

In the definition of "Topology preserving dimension reduction", it requires the data set is sampled from a d-dimensional compact manifold with d<<m. This is too ideal and the assumption often does not hold. For example, when the data has m-dimensional noise, then the so-called invertibility does not exist.

From Fig.4, I don't see any advantage of using i-ML-Enc for visualization. Clearly the classes are more mixed by using the proposed method.

No need to refer to the LIS paper. The LIS loss is just about local distance preserving, which is well known in Multidimensional Scaling (MDS) literature.

It is unknown how to determine the hyperparameters, e.g., gamma^{(l)} and mu^{(l)}.

It is unknown how to determine network architecture for various data types.

The writing is fairly good. The main problem is that the paper is not self-contained. Unless the readers dive into the references, it is hard to understand the methodology, especially many symbols and acronyms.
Some detailed comments:
* All equations are not numbered.
* Some presentation is vague and unexplained. For example, what do dash lines mean in Fig.1? At the end of Section 2, what do you mean "their performance is not impresssive"?
* I cannot find the definitions of d_X and d_Z in the LIS-loss

---

> ### Author Response · Authors · 2020-11-15
> **Reply 1 (R1 Q1)**
>
> Hi, thanks for your pointing out problems in the manuscript. We will update them in the first revision. But we think you might have some misunderstandings, and we answer your questions as following:
> #### R1 Q1: The problem in attack is unclear, and experiment results seem not support invertible dimension reduction (DR).
> Please note that this paper is not to design a new nonlinear dimension reduction (NLDR) method, but to bridge the gap between theoretical manifold learning and the practical DR problem. The main problem under discussion is information-lossless (invertible) DR with the manifold assumption (paragraph 2, 3 in Sec 1), which has not been well-defined by previous DR methods. And we design a simple framework (as figure 2) which is instantiated as i-ML-Enc, and perform empirical studies. The logical relation of each chapter is as follows:
>  - Sec 1. Introduction. Raise the problem of information-lossless NLDR. With the manifold assumption, we give the definition and notations of invertible DR theatrically, in which preserves local geometric and topological structure.
>  - Sec 2. Related work. Provide the background of manifold learning DR methods and the background of two types of inverse models (AE-based reconstruction models and bijective models) for comparison.  Provide theories of compressed sensing as known invertible linear DR.
>  - Sec 3. Method. Sec 3.1 proposes the inverse of canonical embedding as DR process, which can be divided into two stages (inv-ML). Sec 3.2 discuss the information loss between lowest embedding dimension s and different target dimension s’. Sec 3.3 proposes a network implementation (i-ML-Enc) to verify the ideas in Sec 3.1 and Sec 3.2.
>  - Sec 4. Experiment. Test on synthetic datasets and real-world benchmarks which meet the manifold assumption. Sec 4.1 performs comparison experiments with SOTA DR and inverse methods to show that i-ML-Enc has achieved bijective DR (to s-dimension) in the first L-1 layer, and achieved good DR when s>s’ in the L-th layer with information loss. Based on Sec 4.1, we can verify the local geometric and topological structure of learned data manifolds by linear interpolation studies on the L-1-th of i-ML-Enc. Sec 4.3 is an ablation study of i-ML-Enc.
>
> Thus, the proposed i-ML-Enc is only a sub-optimal implementation of the proposed invertible manifold learning process (inv-ML) in this manuscript, which is used to support our empirical studies. Actually, an invertible mapping is not hard to learn, for example, the neural network can be an identity mapping. However, we aimed to learn the constraints of the data manifold in the raw input space, thus the reconstruct loss is non-zero, due to the computation error and the sampling noise leading the data manifold not to lie on the manifold strictly.

---

> > ### Author Response · Authors · 2020-11-15
> > **Reply 1 (R1 Q2-Q4)**
> >
> > #### R1 Q2: Really invertible in experiments？The experimental assumptions are too harsh for practical scenarios?
> > The problem settings are based on manifold learning DR assumption, i.e. points in raw input space are sampled (with noise or not) uniformly on a manifold and the local neighborhood can be regarded as homeomorphic to a Euclidean space, with high plausibility to use Euclidean distances approximating the geodesic distance for preserving the local geometry, which is a basic assumption of manifold learning. Thus, metrics for manifold learning are defined based on comparisons of the local neighborhood of the input space and the latent space. In the paragraph 2 in Sec 1, we try to explain the inverse process of manifold DR by theoretically assuming that the discrete data set X is sampled from the compact data manifold M_0, which is easy to satisfy because all the data points obtained in reality are sampled in a bounded closed region, so the ‘compatibility’ is easy to satisfy.
> >
> > As for the reason for using such an ideal assumption, it’s because we have no prior knowledge of the sampling methods. We can only try to learn the data manifold in the given data set X, rather than remove these non-uniform sampling noises. Practically, we can ignore the effects of the noise when the DR method is robust to it, e.g. neural network based methods ML-Enc and i-ML-Enc. As for experiment datasets, the synthetic datasets can be evaluated by metrics and visualization comparing to the synthetic manifold, e.g. Swiss roll. As for real-world datasets, it’s reasonable to measure the learned embedding by the local neighborhood of the input space even though there is some sampling noise in the dataset.
> >
> > As for inverse abilities of i-ML-Enc, we compare the reconstruction results from L-1-th of i-ML-Enc with other AE-based and bijective SOTA inverse models, where i-ML-Enc shows the lowest RMSE and MSE loss. And we display reconstruction results in figure 4 and A.2.
> >
> > As for DR results visualized in figure 4 and A.2, you should view them from the perspective of manifold learning, rather than clustering or data visualization. The classification measures (Acc) aim to prove that the learned embedding mitigates the ‘dimensionality curse’ (classification on low dimension will be easier on high dimension) while preserving as much information of the input space as possible. Similarly, the visualization provided is for an empirical manifold structure visualization. For example, in figure 4 (d), the compared DR methods, e.g. tSNE and ML-Enc, apparently drop topological information between sub-manifolds in MNIST dataset, while i-ML-Enc preserves it (which seems more mixed). It is our fault for not discussing these results in detail.
> >
> > #### R1 Q3：The representation (details, definitions, and captions) of the paper needs to be improved, which is not self-contained.
> > We will solve these representation problems in the first revision, including complete the definition of loss functions, enrich the caption of the figures, etc. Besides, we refer to LIS loss only because we inherit the network implementation of LIS.
> >
> > #### R1 Q4：Some details, such as the network structures and hyperparameters of i-ML-Enc.
> > We have introduced the network structures of different in A.2. For example, we use the 8 layers MLP as figure 2, which has 7 layers 784D-784D equal-dimensional non-linear layers as the input dimension of MNIST and have a linear transformation of 784D-10D in the 8-th layer. We have released source code and results in the supplement material, and we will add a more detailed description of hyperparameter values in the first revision.

---

> > > ### Comment · AnonReviewer1 · 2020-11-16
> > > **Wrong assumption of the research**
> > >
> > > No prior of sampling noise does not justify the ideal assumption. Oversimplifying the problem gives only over-restricted applications.

---

> > > > ### Author Response · Authors · 2020-11-16
> > > > **Reply to "Wrong assumption of the research"**
> > > >
> > > > Hi, we believe you might misunderstand our assumption of "no prior of sampling noise". We mean that when the DR method is very robust to sampling noise, what we can do is to learn the data manifold from the given data rather than remove the noise. Thus, we ignore the noise when we theoretically discuss the problem of dimension reduction (DR) in Sec 1 and Sec 3.1. To the best of our knowledge, the baseline ML-Enc [1] is very robust to sampling noise.
> > > >
> > > > Two ways to tackle sampling noise in manifold-based DR methods. The first way is to mention sampling noise in the definition of DR, but not explicitly addressing it in their method, which commonly appears in classical methods, e.g. LTSA [2]. The second way is to mention the problem of sampling noise and try to deal with it. For example, [3] tries to subsample data to reduce effects of large noise to some degrees; [4] tries to improve the robustness of classical spectral-based methods by iteratively solving the "repeated eigendirections problem". However, most of them try to make methods more robust to noise rather than remove the noise. As long as the DR method is very robust to sampling noise, we can simply ignore them.
> > > >
> > > > Therefore, it reasonable to ignore the effects of the noise when we discuss DR and verify our ideas by i-ML-Enc. We will explain it in the first revision.
> > > >
> > > > Reference:
> > > >
> > > > [1] Stan Z. Li, et al. Markov-lipschitz deep learning. ArXiv:2006.08256, abs/2006.08256, 2020.
> > > > [2] Zhenyue Zhang et al. Principal manifolds and nonlinear dimensionality reduction via tangent space alignment. SIAM journal on scientific computing, 26(1):313–338, 2004.
> > > > [3] James McQueen, et al. Nearly Isometric Embedding by Relaxation. In Proceedings of NIPS, 2016.
> > > > [4] Daniel Ting, et al. Manifold Learning via Manifold Deflation. 2020. ArXiv:2007.03315v1, abs/2007.03315, 2020.

---

> > ### Comment · AnonReviewer1 · 2020-11-16
> > **The arguments are very weak and not acceptable**
> >
> > Please notice that "invertible" by common understanding is a yes/no problem. If the embedding is only partially invertible, there is nothing new over other approximation methods. So the claim "... the proposed inv-ML not only achieves \emph{better invertible} NLDR in comparison with typical existing methods..." contradicts its so-called main selling point.
> >
> > On the other hand, completely invertible NLDR imposes very strong restrictions on the data and the applications. The paper does not gives any impressive realistic scenarios that perfect recovery can be yielded.

---

> > > ### Author Response · Authors · 2020-11-16
> > > **Reply to "The arguments are very weak and not acceptable"**
> > >
> > > Hi, it's known to all that there are two types of invertilble models (yes or no), i.e. achieve bijective (one-to-one mapping) or not. However, the numerical error is inevitable. A good example to illustrate it is i-RevNet, which is achieved by a one-to-one mapping (equi-dimensional) and has little reconstruction loss. We also achieve invertible NLDR via a one-to-one mapping in the first L-1 layers of i-ML-Enc and have less reconstruction loss than i-RevNet, so we can call it "invertible". We agree with you that the statement in the abstract seems to be contradictory, and we will correct it as "... the proposed inv-ML not only achieves invertible NLDR but also ...".
> > > As for your doubt of the usage of completely invertible NLDR in real scenarios, we have two reasons as following:
> > >  - Completely invertible (only bijective) is not a very strong restriction that will not cause a bad NLDR. Please refer to invertible ResNet [1] and i-RevNet [2] for the following conclusions. (1) If we only add a one-to-one mapping constraint to a network (like invertible ResNet with Lipschitz norm < 1.0), it will not affect its performance. Invertible ResNet and i-RevNet have achieved the same classification accuracy as vanilla ResNet and RevNet. It's also true for i-ML-Enc as shown in Sec 4.1. (2) If we force the network with the one-to-one constrain to be a generative model, e.g. Flow and generative invertible ResNet, it will get bad performance of downstream tasks. These conclusions can also be found in [3].
> > >  - The main purpose of the paper is to study invertible NLDR, the plausibility, and reliability of tangent space estimated by the local neighborhoods which most manifold learning is based on, and to propose a new idea on how to compress the dimension little by little without destroying the topological and geometric structure of the raw data.
> > >
> > > Although we have not proposed a sufficiently optimized invertible NLDR methods, we believe that this problem is worthwhile to be solved. And we will discuss some related problems in our future work, e.g. whether a dimension reduction method should be invertible?
> > >
> > > Reference:
> > >
> > > [1] Jens Behrmann, et al. Invertible Residual Networks. In Proceedings of ICML, 2019.
> > > [2] Jo ̈rn-Henrik Jacobsen, et al. i-RevNet: Deep invertible networks. In Proceedings of ICLR, 2018.
> > > [3] Lynton Ardizzone, et al. Analyzing inverse problems with invertible neural networks. In Proceedings of ICLR, 2019.

---

### Official Review · AnonReviewer2 · 2020-10-28
**Presents a novel method for an invertible non-linear dimensionality reduction with strong empirical results.**

**Rating:** 8
**Confidence:** 4

**Review:**

Summary:
Paper presents a novel encoder-decoder framework for invertible dimensionality reduction. It is composed of multiple stages of homeomorphic embedding, sparse representation, linear compression and an inverse (reconstruction) process to learn invertible non-linear representations. Proposed idea is indeed novel and interesting actualization of geometry preserving dimension reduction shown in Figure 1.

Strengths:
Proposed methods combines multiple ideas work on structure-preserving manifold learning, invertible and distance-preserving sparse representation learning.
Each of the steps above are achieved by NN structure and novel loss functions that impose orthogonality, sparsity and isometry constraints and so on.
Empirical results on synthetic and real-world datasets support the approach and shown efficacy of the method. Ablation studies on adding different components show need of each aspect.

Weakness:
Invertible mapping learned is computed explicitly but can also be learned end-to-end during training. Is there a reason why the prior is preferred.
Paper doesn't not address convergence aspect of the training and how it affects empirical results. Sparsity count (s) for representation is still heuristic and choice is not obvious. Can the RIP property provide a lower bound for choosing s.

Recommendation:
Paper is a clear accept as it introduces a novel method achieve Figure 1 using NN. It is well written and technically sound and qualitative results demonstrate effectiveness of the technique in terms of SOTA results.

---

> ### Author Response · Authors · 2020-11-15
> **Reply 4**
>
> Hi, thank you for your approval of our work and detailed suggestions for revision. We will improve these weaknesses you pointed out in the first revision. And we answer your questions as follows:
> #### R4 Q1:
> There are two sections discussing prior knowledge. The first is the data points sampled from a manifold and represented in the Euclidean coordinate forms. This is the main assumption manifold learning is based on, so we take it as a premise of our research. The second is that when we want to reduce the dimension lower than its intrinsic dimensions s, the neural network will force the manifold to be split, whose topological structure will be destroyed, while the local geometric structure is easier to preserve. After this process, the manifold will be a hyper-plane, so we take it as a prior knowledge for excessive dimension reduction. An example which is easy to imagine is that there is a sphere $S^2$, whose lowest embedding dimension into Euclidean space is 3, and if we force it to be reduced into 2, it will find a point p where the data is not sampled on, and projected the $S^2/{p}$ into a 2-dimensional plane. Therefore, the second prior knowledge of introducing Euclidean space is reasonable to certain degree.
> #### R4 Q2:
> The proposed i-ML-Enc is a simple implementation of inv-ML to validate invertible nonlinear dimension reduction (NLDR). We will add a more detailed discussion in A.2 of training and hyperparameters.
>
> We aim to find an information-lossless nonlinear dimension reduction (NLDR) method based on manifold learning, which is characterized on different datasets, instead of proposing a SOTA method for representation learning. Therefore, the complicated parameter setting is mostly decided by the neural network expressive ability and complexity of datasets. We will add a more detailed discussion in A.2 of training and hyperparameters.

---

### Official Review · AnonReviewer4 · 2020-10-28

**Rating:** 4
**Confidence:** 4

**Review:**

In this paper, the authors propose a novel manifold learning method, via adding a locally isometric smoothness constraint, which preserves topological and geometric properties of data manifold. Empirical results demonstrate the efficacy of their approach. The authors also show that the reliability of tangent space approximated by its local neighborhood is essential to the success of manifold learning approaches.

Overall I found the ideas in the paper somewhat interesting. Many aspects were unclear to me (refer below).

1) In the abstract, the authors claim that in their first step there is no loss of topological information. How did the authors measure this i.e. the sparse coordinate transformation step does not result in any loss of topological information ? Do the results in the paper demonstrate this ?

2) I was not clear to me as to how the authors in their approach are preserving topology and in general geometry of the space while reducing the dimension of the space simultaneously. I would like the authors to add a section on this and/or improve the clarity of their presentation.

3) What is the intuition for the two step process of their approach ? It definitely fits the encoder-decoder framework but is there some other reasoning behind this approach ?

4) In section 3.3, the authors introduce the orthogonal loss wherein they force the weight matrices to be orthogonal. After training, are the weights orthogonal or how close to orthogonal are the weights ? Can we possibly add some other constraint in place of this ?

5) Given the complexity of the model and the different loss functions associated with the objective, how much overhead is involved in model training and execution. Do the authors plan to share any time complexity results so that we can compare training/execution times with other state-of-the-art models ?

6) In the appendix section A.1, what is the intuition behind the exact mathematical form of the expressions used for the Trust and Cont metrics ? I was curious with regards to the exact mathematical expressions used i.e. the different constants and factors involved.

7) I did not quite understand the notation of Figure 2. What is the meaning of the different colored arrows and boxes indicate and how are they used ? There is not discussion on this as well. Similarly for Figure 4 for which there is no analysis or discussion. I would like the authors to include these without which the notation is cumbersome to follow. It is surprising that there is no additional discussion or analysis of the different figures included in the paper, given the captions are not sufficient by themselves.

8) How are the authors deciding on the values of the different hyper-parameters i.e. gamma and alpha in the Appendix ? Overall it felt very heuristic and I did not quite understand how the authors decided on the values used.

9) Some of the loss based definitions in Section 3.3 are unclear to me. In case the authors used some standard/other loss functions from some paper, they could have added those definitions in the main paper or the Appendix etc. I found that the paper was not self contained as such and I had to navigate to the references quite a bit.

I found quite a few typos in the submitted draft. Kindly proofread and correct these. Overall I felt that the paper does not quite achieve lossless compression as well as there is only marginal improvement in terms of results when compared against other approaches. The approach presented in the paper in not clear in many parts.

---

> ### Author Response · Authors · 2020-11-15
> **Reply 2 (R2 Q1-Q9)**
>
> Hi, thanks for your detailed comments on this manuscript, and we will correct these defects in the first revision. But we think you misunderstand that this manuscript proposes a new nonlinear dimension reduction (NLDR) method. Actually, we try to define (in paragraph 2, 3 in Sec 1) and empirically study (Sec 3 and Sec 4) the information-lossless (invertible) dimension reduction (DR) with the manifold assumption. You can refer to the content of the paper in Reply 1 Q1. We answer your questions as follows:
> #### R2 Q1 & Q2:
> We have explained the way we preserve topological structure which is invariant property under homeomorphisms in Sec 3.1, i.e. find a homeomorphism transformation as DR which is equal to a bijective transformation. And if the neural network is an ideal bijective mapping, it can reconstruct the raw data points without any information loss. We verify the invertible abilities in Sec 4.1 by comparing i-ML-Enc to SOTA bijective models i-RevNet, etc.
> #### R2 Q3:
> The two-stage process is discussed in Sec 3.1 where we define the NLDR process as the inverse process of a canonical embedding of the data manifold, thus we can achieve NLDR by performing NLDR in a cascade of several equal-dimensional neural networks (stage 1) and projected the s-sparse results to a lower dimension (stage 2).
> #### R2 Q4:
> After training, we find that the weight of the first L-1-th layers is near-orthogonal because we force the weight matrixes to be orthogonal by soft constraint. It’s possible to achieve inverse in the first L-1-th layers by other invertible constraints.
> #### R2 Q5:
> We think the proposed implementation i-ML-Enc is not the main purpose of this paper, though we achieve SOTA in method comparison. We will add more discussion of training details in A.2 in the first revision.
> #### R2 Q6:
> Trustworthiness (Trust) and Continuity (Cont) are two metrics usually used to measure the latent space representation, which is originally proposed by [1]. Trust checks to what extent the k nearest neighbors of a point are preserved when going from the input space to the latent space, and Cont checks to what extent neighbors are preserved when going from the latent space to the input space. All of these metrics are defined based on comparisons of the input space and the latent space.
> #### R2 Q7:
> We apologize for not giving a more specific explanation and captions of figures in the manuscript. We will make up for this problem in the first revision.
> In Figure 2, the blue line represents linear DR in (b)(c), the red lines in (a) represents nonlinear homeomorphic transformation, the green dash box of “Encoder” represents the NLDR process while the purple dash box represents its inverse process in the decoder.
> In Figure 4, we show the NLDR and reconstruction process of i-ML-Enc by Swiss roll in (a); we show bad NLDR results by compare (b)(c),  it’s known to all that 100D spheres ($S^{100}$) are only possible to embed into $R^{101}$ while 10D half-spheres are possible to embed into $R^{10}$, thus we find i-ML-Enc just projects $S^{100}$ into 2D (unable to perform NLDR) while achieving reasonable results when reduces 10D half-spheres embedded in 100D to 10D.
> #### R2 Q8:
> We have released source code and results in the supplement material, and we will add a more detailed description of hyperparameter values in the first revision.
> #### R2 Q9:
> We will rewrite the loss function and definitions borrowed from other papers, e.g. LIS loss and push-away loss in the first revision.
>
> Reference:
>
> [1] Venna, J. and Kaski, S. Visualizing gene interaction graphs with local multidimensional scaling. In Proceedings of the 14th European Symposium on Artificial Neural Networks, pp. 557–562. d-side group, 2006.

---

> > ### Comment · AnonReviewer4 · 2020-11-18
> > **Response**
> >
> > I would like to thank the authors for their response. I am looking forward to review the revised version of the paper which the authors plan to submit.

---

### Official Review · AnonReviewer3 · 2020-10-31
**This paper proposes an invertible manifold learning (inv-ML) method. It first uses a homeomorphic sparse coordinate transformation to find a low-dimensional representation without loss of topological information. Second, a linear compression is performed on the learned sparse codding to get a trade-off between the target dimension and the incurred information loss. Experiments are conducted on seven datasets to evaluate the reconstruction performance and the generalization ability.**

**Rating:** 5
**Confidence:** 4

**Review:**

Pros:
Figure 4. (a) is informative, which clearly illustrates the invertible learning process.

The authors display the failure cases in Fig 4. (b), which is helpful for other researchers working on this filed. However, more analyses and discussion regarding the failure cases are suggested.

My major concerns are listed as follows.

1, the contribution of this paper seems to be over-claimed. This paper attempts to learn an NLDR without loss of information; however, it is clear that the method in this paper also loses some information in the second step. Moreover, this paper seems to be a combination of previous works (LIS + sparse coordinate transformation), and thus it is important to clearly state the real contributions.

2, the presentation of this paper needs to be improved as many technical details are omitted. For example, in introduction, authors do not well describe how to preserve topology and geometry. This is very important as the preservation of topological and geometric properties of complex structures is a very difficult task [1], while at the same time, low-dimensional structures usually have sophisticated geometric and topological structures [2].

[1] Cohen, et al. "A general theory of equivariant CNNs on homogeneous spaces." NIPS. 2019.
[2] Wakin, et al. “The multiscale structure of non-differentiable image manifolds”. 	In Proceedings o fSPIE, the International Society for Optical Engineering, pages 59141B–1, 	2005.

3, The network mentioned in Fig. 2 is not clear. For example, what is the main difference between the blue and red arrows, the difference between the solid lines and the dash lines, and the difference between the gray rectangles and the white rectangles? Moreover, as this network is based on ML-Enc, without introducing the structure of ML-Enc, it is hard to clearly illustrate the network structure of the proposed method.

4, Figure 4 is important as it visualizes the embeddings of different methods; however, there isn’t any discussion related to figure 4. In addition, in figures 4 (c)-(d), I cannot see the advantage of the proposed model over the comparted ones.

5, There is an interesting phenomenon that the L-1 th layer preserves the most NLDR results, which should be elaborated.


6. there are some typos in this paper, e.g., a NLDR -> an NLDR, and the image quality like figure 1 can be enhanced.

---

> ### Author Response · Authors · 2020-11-15
> **Reply 3 (R3 Q1-Q6)**
>
> Hi, thanks for your detailed comments to point out the defects of the paper, and we will correct them in the first revision. We answer your questions as follows:
> #### R3 Q1:
> The main purpose of this paper is to define (Sec 1) and empirically study (Sec 3 and Sec 4) the information-lossless (invertible) nonlinear dimension reduction (NLDR) with the manifold assumption. That is the gap between mathematical manifold learning and practical DR. Thus, i-ML-Enc is only a replaceable implementation entirely based on the proposed inv-ML.
>
> As we discussed in Sec 4.1 and A.4, i-ML-Enc has achieved invertible NLDR on first L-1 layers, and unreconstructible DR in the L-th layer when s’<s. For example, i-ML-Enc realizes information-lossless NLDR of MNIST from 784D to 125D in the first 7 layers. Low dimension embedding with lossless information is easy to achieve, such as decomposing the 7-th latent space (784D) to 125D even with PCA. But what we want to argue is that the lowest embedding dimension s of MNIST (without information loss) is between 14D and 125D, and thus in the (L-1)-th layer, we reduce it without loss of information. But when further compression of dimension to 10 or even 2 for visualization, it’s reasonable to lose information.
> #### R3 Q2:
> We will adjust the representation in the first revision. But the purpose of Sec 1 is to raise and define the invertible NLDR problem, rather than introduce a new invertible NLDR method. Therefore, we discuss how to preserve topology and geometry in Sec 3.1. Please refer to the content of this paper in Reply 1, R1 Q1.
> #### R3 Q3:
> We will revise a more detailed caption for figure 2. Actually, in figure 2, we show the process to reduce input dimension (blue square) m to the target dimension s’ (the L-th layer), and reconstruct from the lowest embedding dimension s (the L-1-th layer). Specifically, in (b)(c), the blue line represents linear DR and the gray and white patch represents non-zeros and zeros; in (a), the red lines represent nonlinear homeomorphic transformation. Overall, the green dash box of “Encoder” represents the NLDR process, and the purple dash box represents its inverse process in the decoder. We think there is no need to introduce the network structure of ML-Enc, but we will add more details about it in A.2.
> #### R3 Q4:
> We will add more visualization results in A.2 for comparison, and add more detailed commits of figure 4. However, you should view them from the perspective of manifold learning, rather than clustering or data visualization. The visualization provided is for an empirical manifold structure visualization. As for (b)(c) in figure 4, we try to show bad NLDR results of by spheres. Theatrically, 100D spheres ($S^{100}$) are only possible to embed into $R^{101}$ while 10D half-spheres are possible to embed into $R^{10}$, thus we find i-ML-Enc just projects $S^{100}$ into 2D, while achieving reasonable results when reduces 10D half-spheres embedded in 100D to 10D. ML-Enc shows wrong results in (b)(c). As for (d) in figure 4, tSNE and ML-Enc apparently drop topological information between sub-manifolds in MNIST dataset, while i-ML-Enc preserves it. As for (e) in figure 4, COIL-20 dataset is more focused on each sub-manifold of the geometric properties of photographed objects. ML-Enc and tSNE are more focus on preserving the local neighborhood. Thus, i-ML-Enc does achieve a worse result than tSNE on COIL-20.
> #### R3 Q5:
> We design method comparison in Sec 4.1 to prove that i-ML-Enc has achieved invertible NLDR in the L-1-th layer. Therefore, we can further explore the geometric structure of the learned data manifold in the L-1-th layer by the linear interpolation study in Sec 4.2. Additionally, we briefly discuss the s-sparse in the L-1-th layer in A.4.
> #### R3 Q6:
> We will correct these mistakes and enhance picture quality in the first revision.

---

### Author Response · Authors · 2020-11-20
**The first revision in rebuttal**

We sincerely thank reviewers for their detailed comments and repliments. We have submited the first revision to address the defects pointed out by reviewers. The revised parts of the paper are marked in $\textcolor{brown}{brown}$. Specifically, we identify the problem to be addressed and main contributions in this work, the representation is improved, more detailed analysis and results are provided in the experiment section, etc. We are looking forward to your further responses.

---

### Decision · Program_Chairs · 2021-01-07
**Final Decision**

**Decision:**

Reject

**Comment:**


While reviewers find the ideas in the paper interesting, they also raise several major concerns.
In particular, R1 and R4 find the claims of "invertible" and "lossless" to be potentially misleading.
The bijective property is achieve on the first stage (L-1 layers) due to a sequence of one-to-one mappings, as is done in previous work (e.g. i-RevNet)  so the novelty is limited.  As stated by R3,  since the paper is a combination of previous methods, the writing should be substantially improved to clarify what the real, new contributions are. The interpretation of the results (e.g. Figure 4) should also be better explained.